# A novel motif of Rad51 serves as an interaction hub for recombination auxiliary factors

Negar Afshar[1,2], Bilge Argunhan[2]*, Maierdan Palihati[1,2], Goki Taniguchi[1,2], Hideo Tsubouchi[1,2], Hiroshi Iwasaki[1,2]*

[1]School of Life Science and Technology, Tokyo Institute of Technology, Tokyo, Japan; [2]Institute of Innovative Research, Tokyo Institute of Technology, Tokyo, Japan

**Abstract** Homologous recombination (HR) is essential for maintaining genome stability. Although Rad51 is the key protein that drives HR, multiple auxiliary factors interact with Rad51 to potentiate its activity. Here, we present an interdisciplinary characterization of the interactions between Rad51 and these factors. Through structural analysis, we identified an evolutionarily conserved acidic patch of Rad51. The neutralization of this patch completely abolished recombinational DNA repair due to defects in the recruitment of Rad51 to DNA damage sites. This acidic patch was found to be important for the interaction with Rad55-Rad57 and essential for the interaction with Rad52. Furthermore, biochemical reconstitutions demonstrated that neutralization of this acidic patch also impaired the interaction with Rad54, indicating that a single motif is important for the interaction with multiple auxiliary factors. We propose that this patch is a fundamental motif that facilitates interactions with auxiliary factors and is therefore essential for recombinational DNA repair.

*For correspondence:
bargunhan@bio.titech.ac.jp (BA);
hiwasaki@bio.titech.ac.jp (HI)

**Competing interests:** The authors declare that no competing interests exist.

## Introduction

Exogenous factors such as ionizing radiation and genotoxic chemicals can cause DNA damage. However, endogenous processes such as DNA replication and cellular metabolism can also damage DNA (*Lambert and Carr, 2013*). A particularly severe form of DNA damage is a DNA double-strand break (DSB), in which a single normal chromosome is separated into two pathological chromosomes. Homologous recombination (HR) is a major mechanism responsible for accurately repairing DSBs. HR is also critically important for DNA replication (*Ait Saada et al., 2018*). Accordingly, defects in HR lead to genome instability, which drives human diseases such as cancer (*Prakash et al., 2015*).

During HR, the DNA ends that are exposed at a DSB are resected to form 3' overhangs, which are immediately coated by the single-stranded DNA (ssDNA) binding protein RPA (*Symington and Gautier, 2011*). The RecA-family recombinase Rad51 then displaces RPA to form a helical nucleo-protein filament known as the presynaptic filament (*Sun et al., 2020*). This filament can locate a segment of double-stranded DNA (dsDNA) with substantial sequence similarity (i.e.,homology) to the ssDNA (*Greene, 2016*). Upon identifying a homologous region, the Rad51 filament invades the intact dsDNA, displacing the non-complementary strand and forming base pairs with the complementary strand. Within the context of this displacement loop (D-loop) recombination intermediate, the 3'-end of the invading strand can be extended by utilizing the complementary strand as a template for DNA synthesis, allowing for the recovery of lost genetic information (*McVey et al., 2016*). The D-loop can also be expanded by Rad51-driven DNA strand exchange, which increases the extent of base pairing between the two DNA molecules. Consequently, D-loops can be processed to form Holliday junctions, which may be resolved as either crossover or non-crossover products, or

**eLife digest** The DNA molecule contains the chemical instructions necessary for life. Its physical integrity is therefore vital, yet it is also under constant threat from external and internal factors. As a result, organisms have evolved an arsenal of mechanisms to repair damaged DNA. For instance, when the two complementary strands that form the DNA molecule are broken at the same location, the cell triggers a mechanism known as homologous recombination.

A protein known as Rad51 orchestrates this process, helped by an array of other proteins that include Rad55-Rad57, Rad52, and Rad54. These physically bind to Rad51 and activate it in different ways. However, exactly how these interactions take place remained unclear.

To find out more, Afshar et al. examined models of the structure of Rad51, revealing that three of the protein's building blocks create a prominent, negatively charged patch that could be important for DNA repair. Yeast cells were then genetically manipulated to produce a modified version of Rad51 in which the three building blocks were neutralised. These organisms were unable to repair their DNA. Further biochemical tests showed that the modified protein could no longer attach well to Rad55-Rad57 or Rad54, and could not stick to Rad52 at all. In fact, without its negatively charged patch, Rad51 could not find the ends of broken DNA strands, a process which is normally aided by Rad55-Rad57 and Rad52. Taken together, these results suggest that the helper proteins all interact with Rad51 in the same place, even though they play different roles.

Faulty DNA repair processes have been linked to devastating consequences such as cell death or cancer. Understanding the details of DNA repair in yeast can serve as a template for research in more complex organisms, opening the possibility of applications for human health.

they may be disassembled prior to Holliday junction formation, resulting exclusively in non-crossover outcomes (*Mehta and Haber, 2014*).

As the entity capable of identifying homology and driving DNA strand exchange, Rad51 is integral to DNA repair by HR (*Shinohara et al., 1992*; *Muris et al., 1993*; *Sung, 1994*). However, Rad51 does not function alone in vivo. Several other proteins that are required for HR have been identified in the fission yeast *Schizosaccharomyces pombe* including Rad52, Rad54, the Rad51 paralogs Rad55-Rad57, Swi5-Sfr1, and the lesser studied Shu complex (*Ostermann et al., 1993*; *Muris et al., 1996*; *Khasanov et al., 1999*; *Tsutsui et al., 2000*; *Akamatsu et al., 2003*; *Khasanov et al., 2004*; *Martín et al., 2006*). These factors are mostly conserved in the budding yeast *Saccharomyces cerevisiae* despite the large evolutionary distance separating the two yeasts, although it should be noted that the *S. cerevisiae* homolog of Swi5-Sfr1 (Mei5-Sae3) is only involved in meiotic HR (*San Filippo et al., 2008*; *Hoffman et al., 2015*; *Argunhan et al., 2017a*). This suggests that the requirement for a diverse array of auxiliary factors to promote recombinational DNA repair has been conserved throughout evolution, highlighting its importance. However, our understanding of how auxiliary factors promote Rad51 activity remains incomplete, although they seem to perform largely non-overlapping roles (*Zelensky et al., 2014*).

Sfr1 was first identified in *S. pombe* as an interactor of Rad51 that forms a complex with Swi5 specifically involved in promoting Rad51-dependent DNA repair (*Akamatsu et al., 2003*). The Swi5-Sfr1 heterodimer stimulates DNA strand exchange by potentiating Rad51's ATPase activity and stabilizing Rad51 filaments (*Haruta et al., 2006*; *Kurokawa et al., 2008*). In addition to being widely conserved among eukaryotes, the mechanisms through which Swi5-Sfr1 promotes HR appear to be highly similar in yeasts and mammals (*Tsai et al., 2012*; *Su et al., 2014*; *Su et al., 2016*; *Argunhan et al., 2017a*; *Lu et al., 2018*). The Rad51 paralogs Rad55-Rad57 are another group of evolutionarily conserved auxiliary factors. Rad55 and Rad57 were identified in *S. pombe* based on sequence homology and genetic screening, respectively (*Khasanov et al., 1999*; *Tsutsui et al., 2000*). Relatively little is known about the molecular function of Rad55-Rad57 due to the biochemical intractability of the complex, although much like Swi5-Sfr1, it is thought to be an obligate heterodimer. The biochemical analysis that has been performed with *S. cerevisiae* proteins suggests that Rad55-Rad57 promotes Rad51 filament formation on RPA-coated ssDNA and protects the Rad51 filament from disruption by the Srs2 anti-recombinase (*Sung, 1997*; *Liu et al., 2011*). This is consistent with cytological observations in both *S. cerevisiae* and *S. pombe* indicating that the number of DNA

damage-induced Rad51 foci, which represent Rad51 filaments at sites of ongoing DNA repair, are reduced in the absence of Rad55/Rad57 (*Gasior et al., 1998*; *Gasior et al., 2001*; *Akamatsu et al., 2007*).

Among recombination auxiliary factors, the absence of Rad52 results in the most severe phenotype, with deletion mutants displaying DNA damage sensitivity exceeding the *rad51Δ* single mutant; this has been attributed to the absolute dependency of Rad51 on Rad52, as well as Rad51-independent functions of Rad52 (*Doe et al., 2004*). The *rad54Δ* mutant also shows severe DNA damage sensitivity that is indistinguishable from *rad51Δ* (*Muris et al., 1997*), highlighting the absolute requirement for Rad54 in Rad51-dependent DNA repair. By contrast, the *rad57Δ* and *sfr1Δ* mutants show only moderate sensitivity to DNA damage, while the *rad57Δ sfr1Δ* double mutant is as sensitive as the *rad51Δ* single mutant. Based on this additivity, it was proposed that Rad55-Rad57 and Swi5-Sfr1 comprise independent sub-pathways of HR that function in parallel to promote Rad51-dependent DNA repair (*Akamatsu et al., 2003*; *Akamatsu et al., 2007*), although recent evidence has evoked a re-examination of this model (*Argunhan et al., 2020*).

To learn more about the relationship between Rad51 and its auxiliary factors, we sought to identify regions of Rad51 that are important for interactions with auxiliary factors. This led to the identification of an evolutionarily conserved acidic patch comprised of three residues: E205, E206, and D209. Mutation of all three residues to Ala completely ablates Rad51-dependent DNA repair, as does a single charge-reversal mutation, indicating that the negative character of this patch is critical for DNA repair. Mechanistically, these defects in DNA repair stem from abrogation of the interaction with both Rad55-Rad57 and Rad52, leading to impaired recruitment of Rad51 to sites of DNA damage. Remarkably, biochemical reconstitutions indicate that neutralization of the acidic patch also impairs the interaction with Rad54, demonstrating that a single motif of Rad51 is important for its interaction with Rad55-Rad57, Rad52, and Rad54. We propose that this acidic patch of Rad51 comprises a fundamental motif that is essential for interactions with auxiliary factors and therefore recombinational DNA repair.

## Results

### E205, E206, and D209 comprise a protruding acidic patch (PAP) on the exterior of the Rad51 presynaptic filament

Several motifs important for the enzymatic activity of Rad51 are located in the highly conserved ATPase core domain, which is characterized by a β-sheet consisting of mixed parallel and antiparallel β-strands (*Story et al., 1992*; *Pellegrini et al., 2002*; *Shin et al., 2003*; *Conway et al., 2004*). These include the Walker A and B motifs, which are important for ATP binding and hydrolysis (*Saraste et al., 1990*; *Story and Steitz, 1992*), and two DNA binding sites: Site 1, which is comprised of Loop 1 and Loop 2, and Site 2 (*Howard-Flanders et al., 1984*; *Story et al., 1992*). Examination of a homology (i.e., computational) model of *S. pombe* Rad51 (*Sp*Rad51) revealed that the surface of these regions is enriched in positive charge, consistent with roles in the binding of ATP and DNA (*Figure 1A*, left). By contrast, mostly negatively charged regions were found on the opposite face of *Sp*Rad51, including a protruding acidic patch, which we refer to as the PAP hereafter (*Figure 1A*, right). The PAP is among the most negatively charged regions on the surface of *Sp*Rad51 (*Figure 1—figure supplement 1A*) and is situated on a short α-helix preceding the outermost β-strand of the central β-sheet (*Figure 1B* and *Video 1*). Three acidic residues were seen to project out from this α-helix: E205, E206, and D209 (*Figure 1C,D* and *Video 2*).

These residues were also examined in the context of a previously published homology model of the *Sp*Rad51 presynaptic filament (*Ito et al., 2020*) and found to form acidic patches constituting dense negatively charged regions on the exterior of the filament (*Figure 1E* and *Video 3*). The equivalent α-helix in the human Rad51 (*Hs*Rad51) presynaptic filament—the structure of which was determined by cryo-electron microscopy (*Xu et al., 2017*)—also had a negative surface charge (*Figure 1—figure supplement 1B*), as did the corresponding α-helix in the *S. cerevisiae* Rad51 (*Sc*Rad51) presynaptic filament (*Figure 1—figure supplement 1C*)—the structure of which was determined by X-ray crystallography (*Conway et al., 2004*). While E205 was replaced with a conservative Asp residue in *Sc*Rad51 and D209 was conserved in *Hs*Rad51, E206 is the only PAP residue

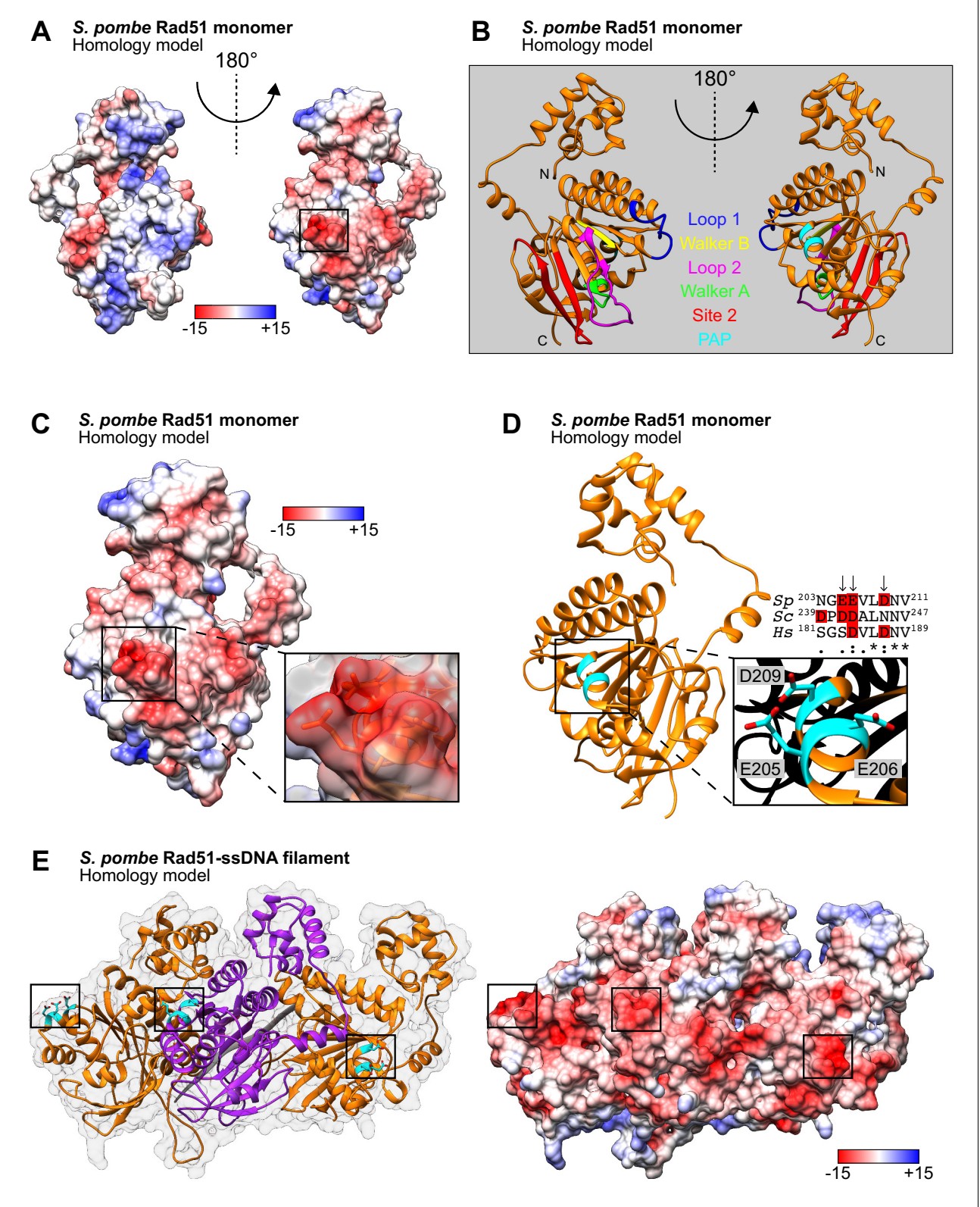

**Figure 1.** E205, E206, and D209 form a protruding acidic patch (PAP) on the exterior of the Rad51 presynaptic filament. (**A**) Homology model of an *Sp*Rad51 monomer (residues 42–360). Surface representation colored according to Coulombic surface charge. The molecule on the left is rotated 180° to visualize the PAP on the right, with the region of interest squared. (**B**) Ribbon depiction of an *Sp*Rad51 monomer with relevant motifs highlighted. The molecules are oriented as in (**A**). (**C**) Surface representation colored according to Coulombic surface charge. The PAP is enlarged with a semi-

*Figure 1 continued on next page*

*Figure 1 continued*

transparent surface revealing residues E205, E206, and D209, which have their side-chains shown. (**D**) Ribbon depiction of *Sp*Rad51 with the α-helix containing the PAP enlarged to illustrate the respective positions of each residue (colored in cyan) with their side-chains revealed (O atoms in red). Sequence alignment shows the corresponding region in *S. pombe*, *S. cerevisiae* and *H. sapiens* (*Sp*, *Sc*, and *Hs*, respectively), with arrows indicating PAP residues in *S. pombe* and acidic residues highlighted in red. (**E**) Ribbon depiction of three *Sp*Rad51 monomers (alternating orange and purple) bound to ssDNA (9-mer poly-dT in gray) with a near-transparent surface for visualization (left). The side-chains of E205, E206, and D209 are revealed in cyan (O atoms in red) and their positions are squared. The surface is made opaque and colored according to Coulombic surface charge to demonstrate that the PAP constitutes dense, negatively charged regions on the exterior of the ssDNA filament (right). Numbers in the legends for (**A,C,E**) are in units of kcal/(mol•*e*).

The online version of this article includes the following figure supplement(s) for figure 1:

**Figure supplement 1.** The PAP is conserved in the *Hs*Rad51 and *Sc*Rad51 presynaptic filaments.

that showed conservation in both *Sc*Rad51 and *Hs*Rad51 (*Figure 1D*). Thus, we initially focused on E206.

## Rad51-E206A is specifically defective in the interaction with Rad55-Rad57

HR plays a particularly important role in the repair of ultraviolet light (UV)-induced DNA damage in *S. pombe* due to the existence of a UV damage endonuclease pathway (*McCready et al., 2000*). To examine whether E206 is important for DNA repair, it was mutated to Ala and a strain containing this mutation at the native locus was constructed. *rad51-E206A* showed the same resistance to UV-induced DNA damage as wild type in a clonogenic survival assay (*Figure 2A*), suggesting that this mutation does not affect the intrinsic ability of Rad51 to repair DNA.

In the *rad57Δ/rad55Δ* background, Rad51-mediated HR is reduced but not abolished, and this remaining recombinational DNA repair is dependent on Swi5-Sfr1. There is a similar reduction in recombinational DNA repair in the *sfr1Δ/swi5Δ* background, where the remaining Rad51-mediated HR is dependent on Rad55-Rad57. However, the *rad57Δ sfr1Δ* double mutant displays a complete loss of Rad51-mediated DNA repair, phenocopying *rad51Δ*. Thus, employing the *rad57Δ* and *sfr1Δ* backgrounds allows for the evaluation of DNA repair promoted by Swi5-Sfr1 and Rad55-Rad57, respectively (*Akamatsu et al., 2003*). The *rad51-E206A rad57Δ* strain was no more sensitive to UV than *rad57Δ* (*Figure 2B*), suggesting that Rad51-E206A is as proficient as wild-type Rad51 in Swi5-Sfr1–dependent DNA repair. By contrast, *rad51-E206A* was as sensitive as *rad51Δ* in the absence of Sfr1 (*Figure 2C*), indicating that the recombinational DNA repair promoted solely by Rad55-Rad57 is ablated by the Rad51-E206A mutation. The generality of these findings was confirmed by performing spot tests with several different genotoxins that induce replication fork stalling and a variety of lesions in DNA (*Figure 2—figure supplement 1A–C*).

The DNA damage sensitivity of *sfr1Δ rad51-E206A* is comparable in severity to *rad51Δ*, which itself shows similar sensitivity to *rad57Δ sfr1Δ* and *rad54Δ* (*Muris et al., 1997*; *Akamatsu et al., 2003*). The *rad52Δ* mutant is even more sensitive to DNA damage, although this added sensitivity stems from Rad51-independent roles of Rad52 (*Doe et al., 2004*); a mutation that abolishes the interaction between Rad51 and Rad52 would be expected to phenocopy *rad51Δ*, not *rad52Δ*. Thus, it is possible that the DNA damage sensitivity associated with *rad51-E206A*, which manifests in the *sfr1Δ* background, reflects a defect in the interaction of Rad51 with Rad52 or Rad54, rather than with Rad55-Rad57. We sought to distinguish between these possibilities genetically. Rqh1 is a RecQ-

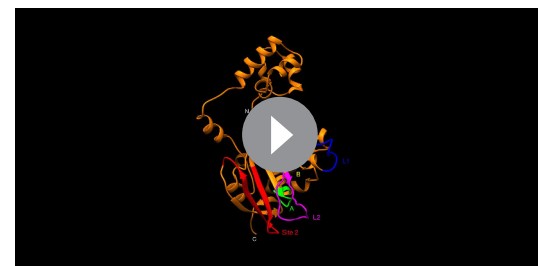

**Video 1.** A model of the *Sp*Rad51 monomer with motifs labeled. A homology model of the *Sp*Rad51 monomer is depicted in ribbon form and labeled as follows: N and C, N- and C-termini; L1 and L2, loop 1 and loop 2 of DNA binding site 1; A and B, Walker A and Walker B motifs; Site 2, DNA binding site 2. Residues E205, E206, and D209, which constitute the protruding acidic patch (PAP), are shown in cyan following 180° rotation of the model in the y-axis.
https://elifesciences.org/articles/64131#video1

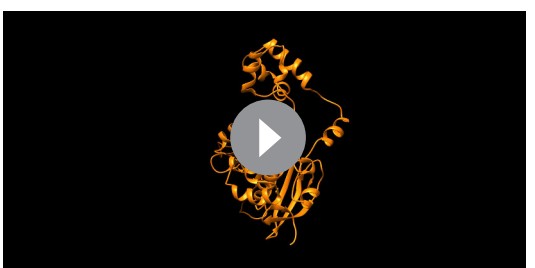

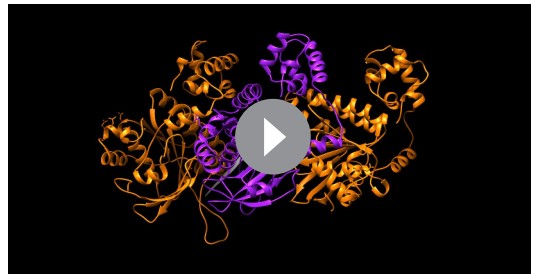

**Video 2.** Visualization of the protruding acidic patch (PAP) in monomer form. A homology model of the *Sp*Rad51 monomer is depicted in ribbon form. The side-chains of residues E205, E206, and D209 are shown, intermittently highlighted in cyan (O atoms in red). The surface is made opaque and colored according to Coulombic surface charge to demonstrate that these residues constitute a dense negatively charged patch that we refer to as the PAP.
https://elifesciences.org/articles/64131#video2

**Video 3.** Visualization of the protruding acidic patch (PAP) in filament form. A homology model of the *Sp*Rad51-ssDNA filament. Ribbon depiction of three monomers (alternating orange and purple) bound to ssDNA (9-mer poly-dT in gray). The side-chains of residues E205, E206, and D209 are shown, intermittently highlighted in cyan. The surface is made opaque and colored according to Coulombic surface charge to demonstrate that these residues constitute dense negatively charged patches on the exterior of the ssDNA filament.
https://elifesciences.org/articles/64131#video3

family helicase (homologous to Sgs1 in *S. cerevisiae* and the BLM and WRN helicases in humans) that functions in S phase to prevent the accumulation of toxic recombination intermediates that arise during DNA replication (*Murray et al., 1997*; *Stewart et al., 1997*). Accordingly, *rqh1Δ* cells are sensitive to the ribonucleotide reductase inhibitor hydroxyurea (HU). It was previously shown that *rad57Δ* and *sfr1Δ* robustly suppress the HU sensitivity of *rqh1Δ* whereas *rad52Δ* and *rad54Δ* do not, presumably because the former mutations reduce HR while the latter mutations eliminate it completely (*Hope et al., 2005*). If Rad51-E206 is defective in the interaction with Rad55-Rad57 rather than Rad52/Rad54, then the *rqh1Δ rad51-E206A* strain should phenocopy *rqh1Δ rad57Δ* rather than *rqh1Δ rad52Δ/rad54Δ*. Consistently, *rad51-E206A* suppressed the HU sensitivity of *rqh1Δ* to almost the same degree as *rad57Δ* (*Figure 2D*). Furthermore, the suppression conferred by *sfr1Δ*, which is dependent on Rad55-Rad57 (*Hope et al., 2005*), was ablated by *rad51-E206A*. By contrast, the suppression imparted by *rad57Δ* was epistatic to *rad51-E206A*, suggesting that they involve the same mechanism. At 5 mM HU, *rad57Δ sfr1Δ* could partially suppress *rqh1Δ* sensitivity, whereas *rad51Δ* and *rad54Δ* could not. Importantly, the *sfr1Δ rad51-E206A* strain was still similar to *rad57Δ sfr1Δ*, strongly suggesting that the defect associated with *rad51-E206A* is related to Rad55-Rad57.

Immunoblotting experiments revealed that the level of Rad51-E206A was comparable to wild-type Rad51 (*Figure 2—figure supplement 1D*), indicating that the DNA repair defect of *rad51-E206A* is not caused by reduced protein stability. Previous yeast two-hybrid (Y2H) analysis suggested that Rad51 physically interacts with Rad55-Rad57 (*Hays et al., 1995*; *Johnson and Symington, 1995*; *Tsutsui et al., 2001*). Thus, a feasible explanation for the DNA damage sensitivity of *rad51-E206A* is that the E206A mutation disrupts the physical interaction between Rad51 and Rad55-Rad57. To test this, a sequence encoding 12 copies of the V5 epitope was fused to *rad55⁺* at its native locus, yielding the *rad55-12xV5* strain. This strain was as resistant to DNA damage as the untagged wild type (*rad55⁺*; *Figure 2—figure supplement 1E*), indicating that the tag does not interfere with the function of Rad55-Rad57 in DNA repair. In vivo co-immunoprecipitation (co-IP) experiments were performed in both *sfr1⁺* and *sfr1Δ* backgrounds. While robust signal was observed for wild-type Rad51, substantially less Rad51-E206A was seen to co-IP with Rad55 (*Figure 2E*), indicating that the E206A mutation impairs Rad51–Rad55-Rad57 complex formation. The presence of Sfr1 did not have an effect on complex formation, irrespective of the E206A mutation. These results suggest that the DNA repair defect associated with *rad51-E206A* is related to impaired Rad51–Rad55-Rad57 complex formation and that the suppression of this DNA damage sensitivity by Swi5-Sfr1 is not through enhancing physical binding.

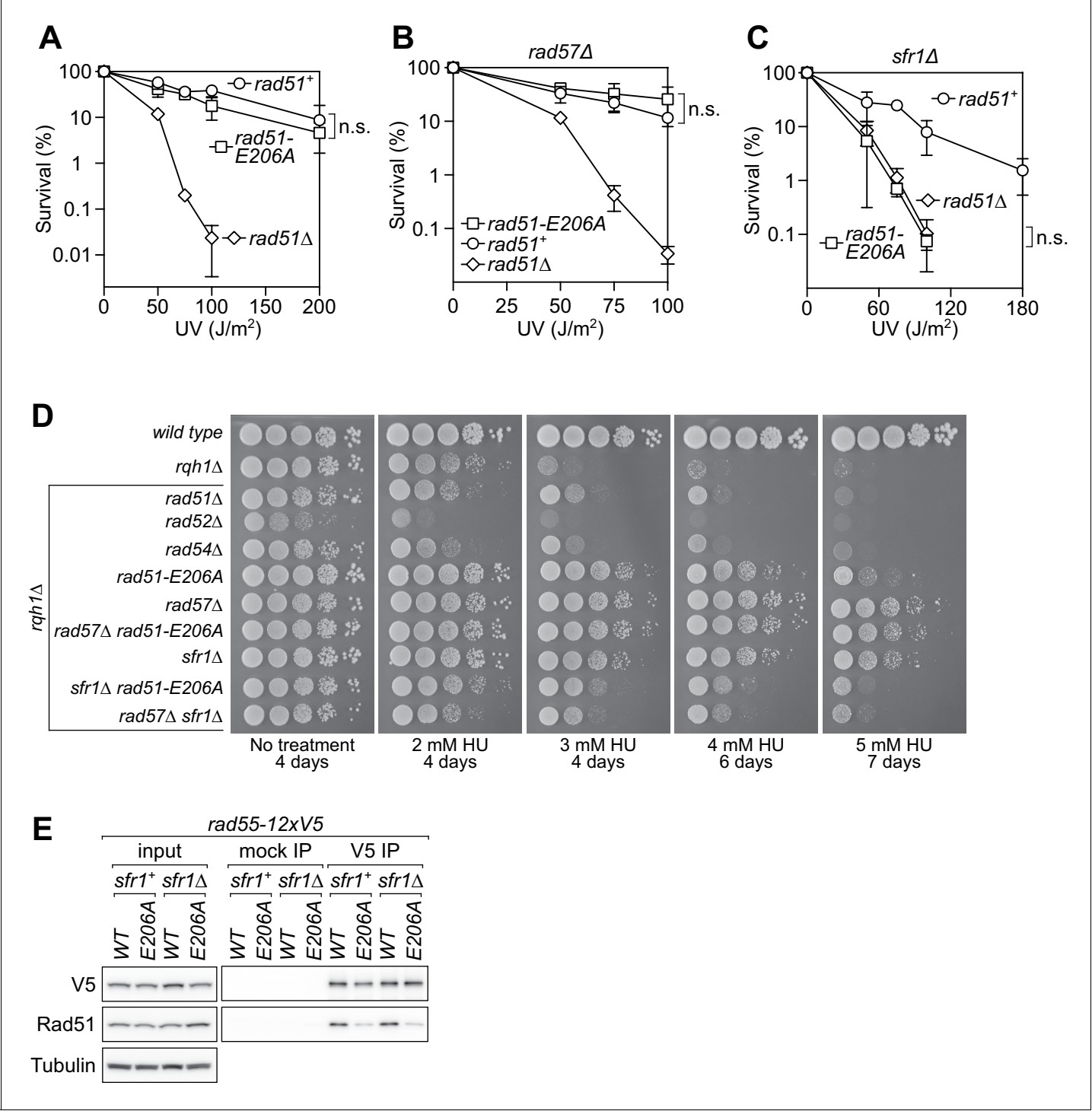

**Figure 2.** The *rad51-E206A* mutant is specifically defective in the interaction with Rad55-Rad57. (A–C) Following acute UV irradiation, a clonogenic assay was employed to test the survival of *rad51*[+], *rad51-E206A*, and *rad51Δ* in the wild-type background (A), the *rad57Δ* background (B), and the *sfr1Δ* background (C). Statistical significance at the highest dose of UV was assessed by unpaired two-tailed t-test. n.s., not significant (A, p=0.506; B, p=0.242; C, p=0.593). (D) Tenfold serial dilutions of the indicated strains were spotted onto standard media without treatment or containing hydroxyurea (HU). Following growth for the indicated time at 30°C, plates were imaged. (E) Soluble cell extracts treated with a benzonase-like nuclease were prepared from each strain under native conditions (input). Immunoprecipitation (IP) was then performed with mock (human IgG from non-immunized animal) or anti-V5 antibodies. Tubulin serves as a loading control. Data in (A–C) are means of three independent experiments and error bars depict standard deviation.

*Figure 2 continued on next page*

*Figure 2 continued*

The online version of this article includes the following source data and figure supplement(s) for figure 2:

**Source data 1.** Survival (%) following UV irradiation for data in *Figure 2A–C*.

**Figure supplement 1.** Specificity of the DNA damage sensitivity of *rad51-E206A*.

## Rad51-E206A retains normal recombinase activity and can be stimulated by Swi5-Sfr1

Although E206 does not belong to a canonical motif involved in ATP hydrolysis or DNA binding, it remained formally possible that the E206A mutation impaired the enzymatic activity of Rad51. Rad51-E206A was therefore purified to homogeneity from *Escherichia coli* to investigate its biochemical properties (*Figure 3—figure supplement 1*).

Rad51-E206A shifted both ssDNA and dsDNA to a similar extent as wild-type Rad51 in electrophoretic mobility shift assays (EMSAs; *Figure 3A,B*), suggesting that the E206A mutation does not affect the ability of Rad51 to bind DNA. In addition to DNA binding, ATP hydrolysis by Rad51 is important for the DNA strand exchange reaction (*Ito et al., 2018*). Furthermore, Swi5-Sfr1 stimulates the ATPase activity of Rad51 (*Haruta et al., 2006*). We therefore examined whether Rad51-E206A is proficient for ATP hydrolysis, both with and without Swi5-Sfr1. In the absence of Swi5-Sfr1, the ATP turnover number $(k_{cat})$ of both Rad51 and Rad51-E206A was ~0.2 min$^{-1}$ (*Figure 3C*). The inclusion of Swi5-Sfr1 elicited an approximately twofold increase in ATP hydrolysis by both proteins, demonstrating that Rad51-E206A can hydrolyze ATP like wild type and is proficient for the ATPase stimulation imparted by Swi5-Sfr1.

Next, an assay with plasmid-sized DNA substrates was employed to examine the strand exchange activity of Rad51-E206A (*Figure 3D*). Rad51 drives the pairing of circular ssDNA (css) with homologous linear dsDNA (lds) to yield joint molecule intermediates (JMs). Following strand transfer over the length of the dsDNA substrate, JMs are converted into nicked-circular dsDNA molecules (NCs), which are the products in this assay. The different DNA species can be separated by agarose gel electrophoresis and visualized. Under our standard reaction conditions, wild-type Rad51 cannot promote JM or NC formation in the absence of auxiliary factors (*Haruta et al., 2006*; *Kurokawa et al., 2008*). However, when reaction conditions were modified by increasing the concentrations of Rad51 and DNA substrates, both Rad51 and Rad51-E206A were seen to drive the efficient pairing of css and lds to produce JMs in the absence of auxiliary factors (*Figure 3E*). This allowed us to evaluate whether the E206A mutation impaired the intrinsic strand exchange activity of Rad51. Although neither protein was able to drive the robust accumulation of NC, we nevertheless quantified the total yield (JM + NC) at each time point and found them to be comparable for both Rad51 and Rad51-E206A.

Our genetic analysis suggested that Rad51-E206A is proficient for the DNA repair promoted by Swi5-Sfr1. To corroborate these findings, strand exchange reactions were supplemented with Swi5-Sfr1. Note that under these standard assay conditions, which differ from those employed above, Rad51 alone cannot promote JM or NC formation, thus allowing us to better examine the effect of Swi5-Sfr1 (*Haruta et al., 2006*; *Kurokawa et al., 2008*). The inclusion of Swi5-Sfr1 efficiently stimulated both Rad51 and Rad51-E206A, with a similar accumulation of JMs and NC observed in both cases (*Figure 3F*). Taken together, these results indicate that Rad51-E206A retains intrinsic recombinase activity and a functional interaction with Swi5-Sfr1.

## The PAP is essential for Rad51-dependent DNA repair

Our results with Rad51-E206A suggested that the PAP is required specifically for the interaction with Rad55-Rad57. We sought to test whether other mutations in the PAP also affect the interaction with Rad55-Rad57. Strains were constructed in which the *rad51*$^+$ gene at its native locus was replaced with *rad51-E205A*, *rad51-D209A*, or *rad51-EED* (E205A, E206A, D209A). Structural models revealed that the negative surface charge of the PAP is completely neutralized by the EED mutation without affecting its protruding nature (*Figure 4—figure supplement 1A,B*).

The DNA damage sensitivity of these mutant strains was assessed by spot-test. Like *rad51-E206A*, the *rad51-E205A* strain did not show any DNA damage sensitivity in the presence of both Rad55-Rad57 and Swi5-Sfr1 (*Figure 4A*). By contrast, *rad51-D209A* showed marked sensitivity to

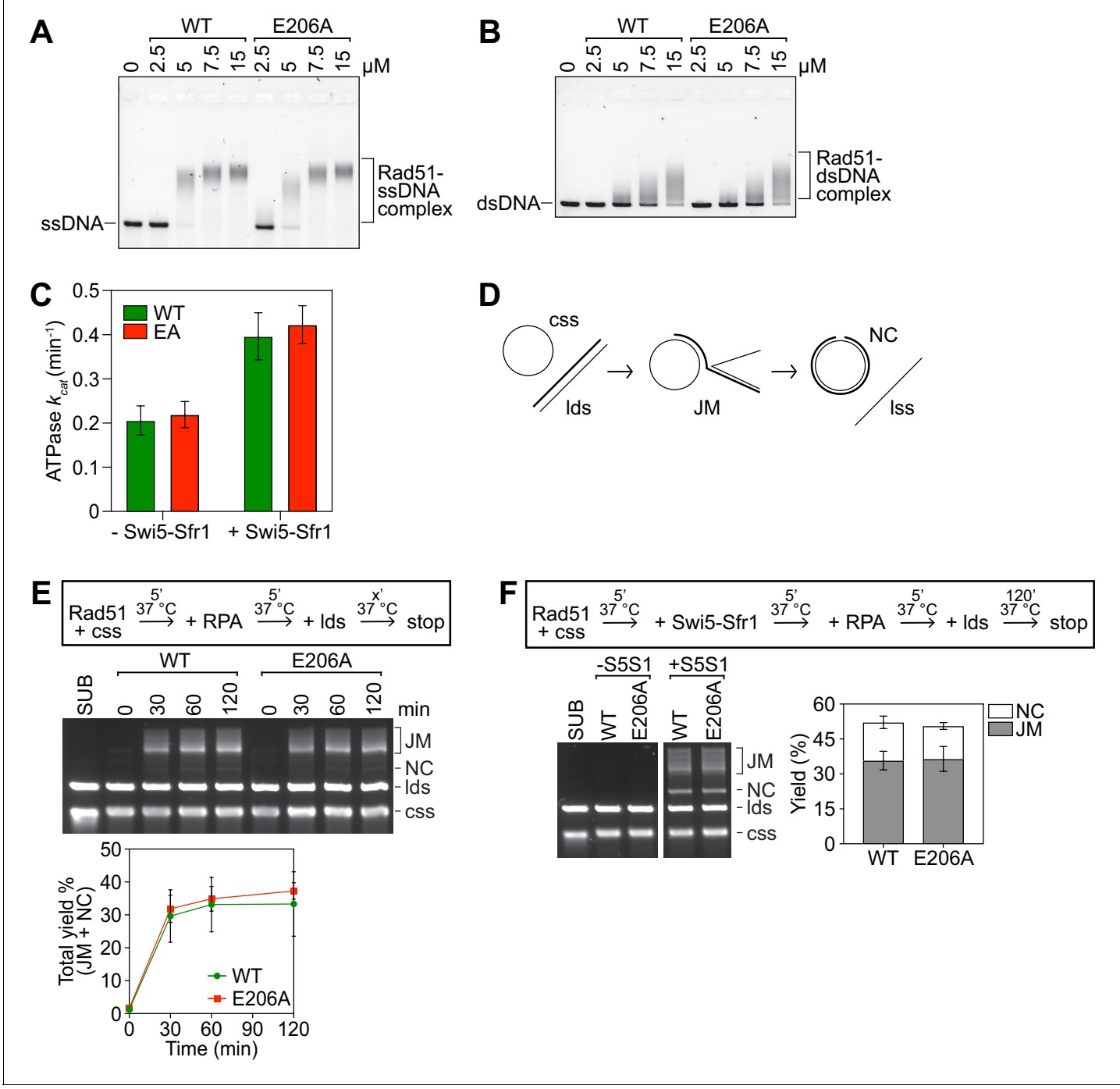

**Figure 3.** Rad51-E206A retains normal DNA binding and recombinase activity. (A,B) The indicated concentrations of Rad51 (WT) or Rad51-E206A (E206A) were incubated with 30 micromolar nucleotide (μM nt) PhiX174 virion DNA (ssDNA; A) or 20 μM nt of linearized PhiX174 RF I DNA (dsDNA; B), protein-DNA complexes were crosslinked with glutaraldehyde and then resolved by agarose gel electrophoresis. (C) The ATPase activity of 5 μM Rad51 (WT) and Rad51-E206A (EA) was measured in the presence of ssDNA (10 μM nt), with or without Swi5-Sfr1 (0.5 μM), and $k_{cat}$ was calculated. (D) Schematic of the strand exchange assay with full-length DNA substrates (PhiX174 virion ssDNA and ApaLI-linearized PhiX174 RF I dsDNA). (E) Strand exchange reactions were conducted according to the scheme outlined above the gel. Rad51 (WT or E206A), 15 μM. RPA, 1 μM. cssDNA, 30 μM nt. ldsDNA, 20 μM nt. (F) Strand exchange reactions were conducted according to the scheme outlined above the gel. Rad51 (WT or E206A), 5 μM. Swi5-Sfr1 (S5S1), 0.5 μM. RPA, 1 μM. cssDNA, 10 μM nt. ldsDNA, 10 μM nt. Data in (C,E,F) are means of three independent experiments and error bars depict standard deviation.

The online version of this article includes the following source data and figure supplement(s) for figure 3:

**Source data 1.** $k_{cat}$ values in *Figure 3C* and strand exchange yield (%) in *Figure 3E,F*.

*Figure 3 continued on next page*

*Figure 3 continued*

**Figure supplement 1.** Purity analysis of Rad51 and Rad51-E206A.

DNA damage, although it was still significantly more resistant than *rad51Δ*. Strikingly, the *rad51-EED* mutant was as sensitive to DNA damage as *rad51Δ*. Since the E205A and E206A mutations alone did not sensitize cells to DNA damage, but combining them with the partially functional D209A completely incapacitated Rad51, we also examined the *rad51-EE* strain, in which both Glu residues are mutated to Ala. Unlike *rad51-E205A* and *rad51-E206A*, *rad51-EE* showed moderate sensitivity to DNA damage, although this was milder than the sensitivity of *rad51-D209A*. These results indicate that neutralization of the PAP completely abolishes Rad51-dependent DNA repair, and while all three residues are important, D209 plays a more prominent role than E205 and E206.

Because our data suggested that the E206A mutation impairs the interaction of Rad51 with Rad55-Rad57, in vivo co-IP experiments were performed to examine how other mutations in the PAP affect complex formation with Rad55-Rad57. A reduction in the amount of Rad51 that co-IPs with Rad55-12xV5 was observed in the *rad51-EE*, *rad51-D209A*, and *rad51-EED* strains (*Figure 4B*), and this reduction roughly correlated with the DNA damage sensitivity of the corresponding mutants. This result suggests that the DNA damage sensitivity of PAP mutants is partly due to reduced complex formation with Rad55-Rad57. However, given that some complex formation was still observed in the *rad51-EED* strain, we infer that the PAP is important but not essential for complex formation with Rad55-Rad57.

In contrast to what was observed in the presence of Sfr1, *rad51-E205A* showed mild DNA damage sensitivity in the *sfr1Δ* background (*Figure 4C*) and this modest difference became more obvious at higher doses of UV irradiation (*Figure 4—figure supplement 1C*). Both the *rad51-EED* and *rad51-EE* mutants showed comparable sensitivity to *rad51Δ* in this background, as was expected from the phenotype of *rad51-E206A*. Notably, the *rad51-D209A* strain was also as sensitive to DNA damage as *rad51Δ* in the absence of Sfr1. These results are consistent with the notion that the PAP is important for DNA repair promoted by Rad55-Rad57. However, because the DNA damage sensitivity of *rad51-EED* is similar to that of *rad51Δ* (*Figure 4A*), which clearly exceeds that of *rad57Δ*, it seemed likely that the function of the PAP is not restricted to Rad55-Rad57–dependent DNA repair. To directly test if the PAP is involved in Rad55-Rad57–independent DNA repair, PAP mutants were introduced into the *rad57Δ* background. The *rad51-E205A* mutant was as resistant to DNA damage as *rad51⁺* in the absence of Rad57, suggesting that Rad55-Rad57–independent DNA repair mechanisms are not impaired in this mutant (*Figure 4D*). By contrast, *rad51-EE* showed a modest increase in sensitivity compared to *rad51⁺*. *rad51-D209A* was even more sensitive than *rad51-EE*, although this sensitivity was still not as severe as *rad51-EED* and *rad51Δ*. Consistently, *rad51-EED* and *rad51Δ* were the only strains among this set that showed a clear slow-growth phenotype (note the small colony size on the 'No treatment' plate in *Figure 4D*). The fact that the D209A and EE mutations further sensitized *rad57Δ* cells to DNA damage, combined with the severe sensitivity of *rad51-EED*, suggests that Rad55-Rad57–independent DNA repair is also impaired in these mutants. These results imply that the PAP is also relevant to the function of auxiliary factors other than Rad55-Rad57 (explored below).

If the negativity of the PAP is important for Rad51-dependent DNA repair, we reasoned that a charge-reversal mutation would be more disruptive than the Asp/Glu-to-Ala mutations employed thus far. The E205A and E206A mutations did not sensitize cells to DNA damage in the presence of both Rad55-Rad57 and Swi5-Sfr1 (*Figure 4A*). However, because E206 shows a higher degree of conservation than E205, it was mutated to Lys, yielding the *rad51-E206K* strain. In stark contrast to the *rad51-E206A* strain, *rad51-E206K* showed a clear slow-growth phenotype and was as sensitive to DNA damage as *rad51Δ* in the presence of both Rad55-Rad57 and Swi5-Sfr1 (*Figure 4—figure supplement 1D*), just like *rad51-EED*. Taken together, these results show that the negative character of the PAP is integral to recombinational DNA repair.

## The PAP is crucial for the recruitment of Rad51 to DNA damage sites

To elucidate the mechanistic defects in DNA repair associated with neutralization of the PAP, previous immunostaining protocols for the visualization of Rad51 foci were adopted (*Loidl and Lorenz,*

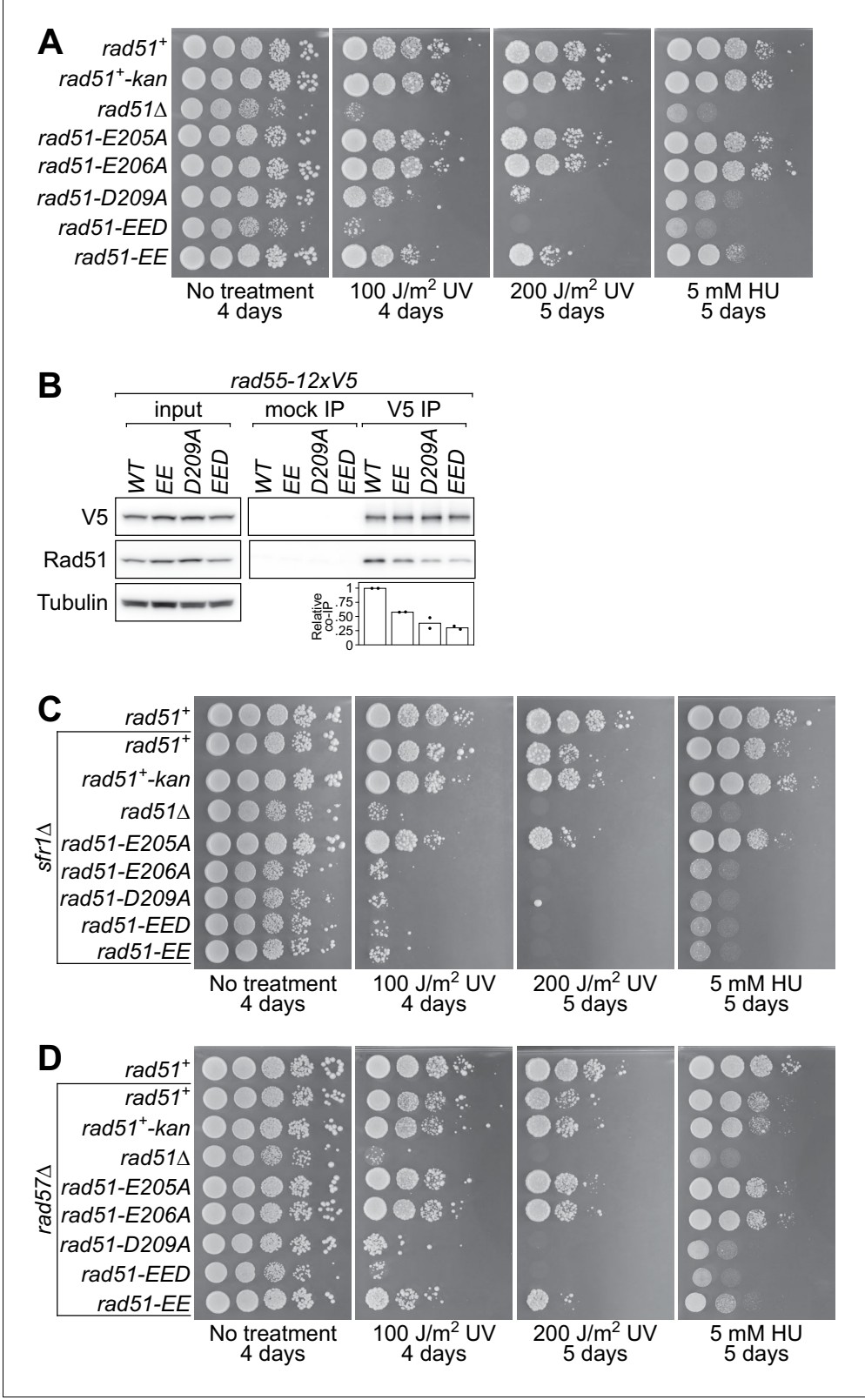

**Figure 4.** The PAP is essential for Rad51-dependent DNA repair. (A,C,D) Tenfold serial dilutions of the indicated strains were spotted onto standard media with or without acute UV irradiation, or standard media containing 5 mM hydroxyurea (HU). Following growth at 30°C for the indicated time, plates were imaged. (B) Soluble cell extracts treated with a benzonase-like nuclease were prepared from each strain under native conditions (input).

*Figure 4 continued on next page*

*Figure 4 continued*

Immunoprecipitation (IP) was then performed with mock (human IgG from non-immunized animal) or anti-V5 antibodies. Tubulin serves as a loading control. For quantification, Rad51 signal was normalized to V5 signal and expressed relative to wild type. Data in (**B**) are the means of two independent biological replicates with individual values shown.

The online version of this article includes the following source data and figure supplement(s) for figure 4:

**Source data 1.** Relative co-IP of Rad51 (%) for *Figure 4B*.

**Figure supplement 1.** The EED mutation neutralizes the PAP of *Sp*Rad51 and is phenocopied by the E206K mutation.

---

*2009*; *Argunhan et al., 2017b*). Surface-spread nuclei were prepared from log-phase cultures, with or without prior UV irradiation, and immunostained with a polyclonal anti-Rad51 antibody (*Figure 5A*). Nuclei were then picked manually and foci were quantified by an automated approach using FIJI software (see Materials and methods for more details; *Schindelin et al., 2012*). Examples of nuclei scored by this method are shown in *Figure 5B*.

In the absence of UV irradiation, the vast majority of nuclei in all strains lacked Rad51 foci (*Figure 5—figure supplement 1*). By contrast, UV-irradiated nuclei contained substantially more Rad51 foci, and this was not seen in the *rad51Δ* strain, confirming the DNA damage-dependence and specificity of these cytological entities (*Figure 5C*). Nuclei from wild-type and *rad52Δ* strains formed an average of 8.6 and 1.7 foci, respectively, in close agreement with a previous report (*Lorenz et al., 2009*). As expected, both *sfr1Δ* and *rad57Δ* strains formed fewer foci than wild type, and *rad57Δ sfr1Δ* formed even fewer foci still (*Akamatsu et al., 2007*). Strikingly, nuclei from *rad51-EED* showed a drastic reduction in the number of foci, much like the *rad57Δ sfr1Δ* double mutant and the *rad52Δ* single mutant. These results indicate that the severe DNA damage sensitivity of *rad51-EED* stems from defects in the mechanisms promoting the recruitment of Rad51 to sites of DNA damage.

## Rad51-EED retains intrinsic recombinase activity and can be stimulated by Swi5-Sfr1

The DNA damage sensitivity and impaired Rad51 recruitment phenotypes of *rad51-EED* are clearly more severe than *rad57Δ*, pointing towards additional roles of the PAP beyond facilitating the interaction with Rad55-Rad57. This could reflect an impairment in the intrinsic activity of Rad51-EED. Alternatively, the EED mutation could impair the interaction with Swi5-Sfr1, since the phenotypes of *rad51-EED* are comparable to *rad57Δ sfr1Δ*. To test these two possibilities, Rad51-EED was purified to homogeneity (*Figure 6—figure supplement 1A*).

Rad51-EED shifted ssDNA and dsDNA comparably to wild-type Rad51 in EMSAs (*Figure 6—figure supplement 1B,C*), suggesting that the EED mutation does not affect the ability of Rad51 to bind DNA. Because reaction products (NCs) were not observed in the assay that was previously employed to monitor the intrinsic strand exchange activity of Rad51 (*Figure 3D,E*), a shorter lds substrate was utilized with the reasoning that this might prove a less challenging substrate (*Figure 6A*). Both Rad51 and Rad51-E206A generated similar amounts of partial duplex molecules (PD), which are the reaction products in this assay (*Figure 6B*). Unexpectedly, Rad51-EED consistently displayed increased strand exchange activity. While the reason for this is unclear, these results at least indicate that neutralization of the PAP does not impair the intrinsic activity of Rad51.

If the EED mutation impairs the interaction with Swi5-Sfr1, then the binding of Swi5-Sfr1 to Rad51-EED should be abrogated. Co-IP experiments with purified proteins revealed reproducible differences in the amount of Rad51-E206A and Rad51-EED that co-IP'd with Sfr1, but these differences were relatively subtle (*Figure 6C*). Moreover, in our canonical strand exchange assay (*Figure 3D*), these differences in physical binding did not affect the stimulation of Rad51 by Swi5-Sfr1 (*Figure 6D*), suggesting that they are not of functional significance. We also examined the interaction of Swi5-Sfr1 with Rad51-E206K. Similarly to Rad51-EED, we saw a reproducible reduction in the co-IP of Rad51-E206K with Sfr1, but this difference is relatively subtle (less than twofold; *Figure 6—figure supplement 1D*). These results indicate that PAP mutations do not impair the potentiation of Rad51 by Swi5-Sfr1, leading us to conclude that the severe DNA damage sensitivity of the *rad51-EED* strain is unlikely to be due to a defect in the interaction with Swi5-Sfr1.

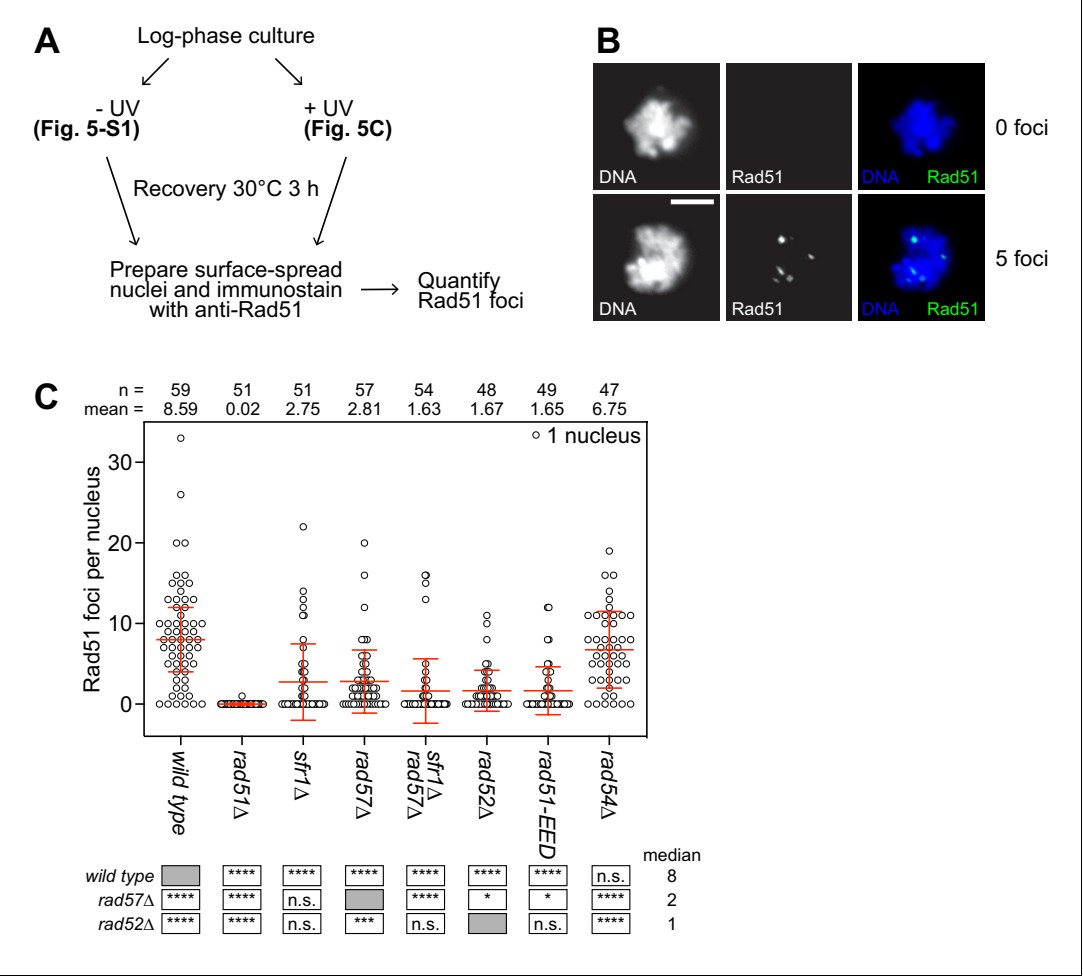

**Figure 5.** The PAP is critical for the recruitment of Rad51 to sites of DNA damage. (**A**) Schematic of cytological analysis. (**B**) Representative images of UV-irradiated nuclear spreads containing 0 or 5 Rad51 foci. Scale bar, 5 μm. (**C**) The indicated strains were grown to log phase and cultures were split into two. One sub-culture was UV-irradiated (200 J/m$^2$; shown here) and the other was not (shown in *Figure 5—figure supplement 1*). Following 3 hr of recovery, nuclei were surface-spread and immunostained with a polyclonal PVDF-purified anti-Rad51 antibody. The indicated number of nuclei were manually picked, images in the DAPI and Rad51 channels were captured, and semi-automated quantification of foci was performed using FIJI software. Bars depict the median and interquartile range. Mean number of foci are shown for each strain. Statistical analysis was by Wilcoxon ranked sum test with the indicated median values. n.s., not significant (p>0.05). * p<0.05. ***p<0.001. ****p<0.0001.

The online version of this article includes the following source data and figure supplement(s) for figure 5:

**Source data 1.** Number of foci per nucleus following UV irradiation for *Figure 5C*.

**Figure supplement 1.** DNA damage dependency of Rad51 foci.

**Figure supplement 1—source data 1.** Number of foci per nucleus without UV irradiation.

## The PAP is important for the interaction of Rad51 with both Rad52 and Rad54

While a defect in the interaction with Rad54 could explain the severity of DNA damage sensitivity observed for *rad51-EED*, this is unlikely to be the cause of the sensitivity as *rad54Δ* is not defective in the recruitment of Rad51 to sites of DNA damage (*Figure 5C*). Thus, the remaining possibility that could account for the phenotypes of *rad51-EED* is that the EED mutation impairs the interaction of Rad51 with Rad52. To examine whether Rad51-Rad52 complex formation was affected by PAP mutations, in vivo co-IP experiments were performed. Comparable amounts of Rad52 were seen to co-IP with Rad51, Rad51-E206A, and Rad51-EE, whereas a clear reduction was observed for Rad51-D209A (*Figure 7A*). Strikingly, Rad52 was completely undetectable in the Rad51-EED

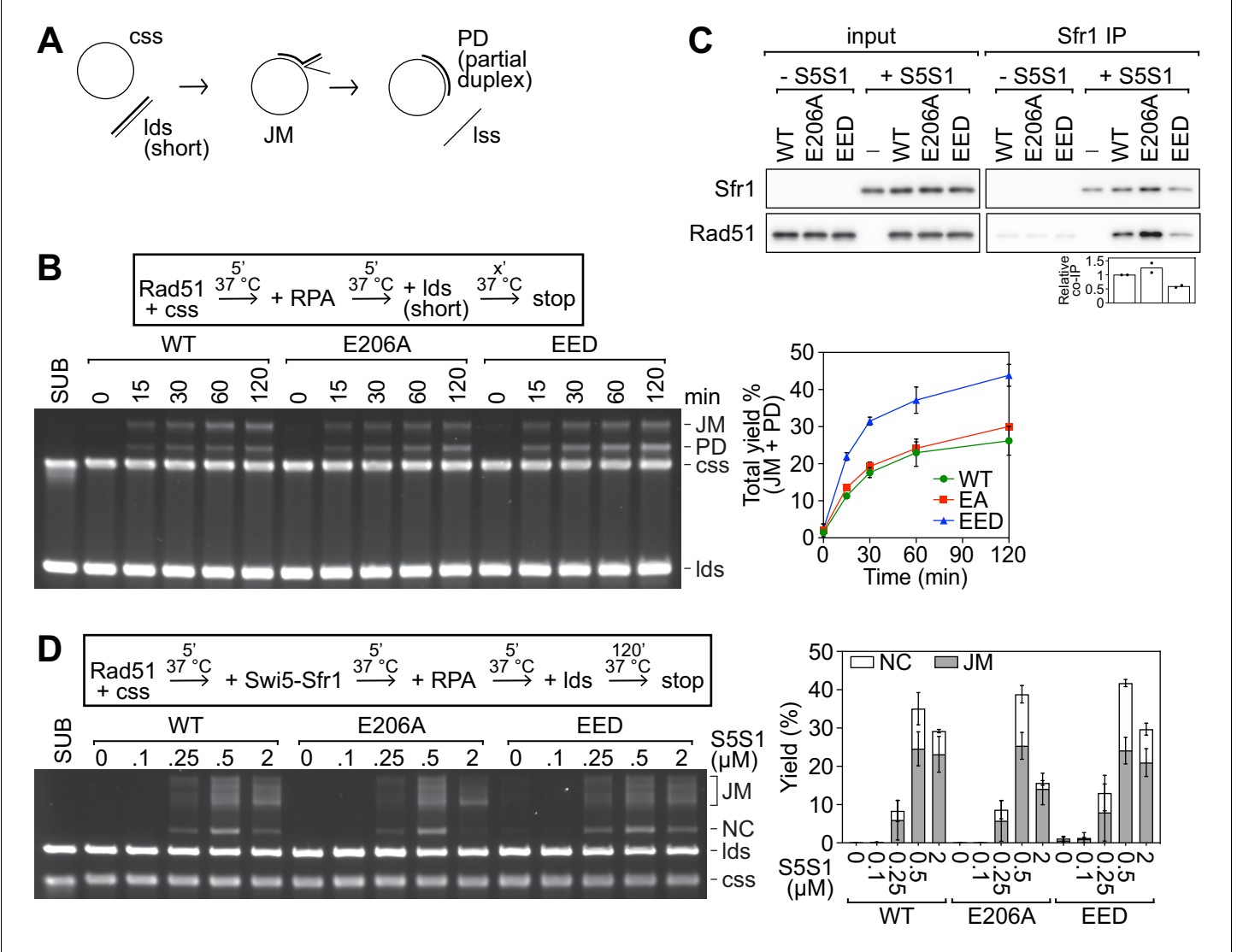

**Figure 6.** Neutralization of the PAP does not impair the intrinsic activity of Rad51 or the stimulation by Swi5-Sfr1. (A) Schematic of the strand exchange assay with shortened DNA substrates (PhiX174 virion ssDNA and a 1.6 kb fragment of PhiX RF I dsDNA). (B) Strand exchange reactions were conducted according to the scheme outlined above the gel. Rad51 (WT, E206A, or EED), 15 µM. RPA, 1 µM. cssDNA, 30 µM nt. ldsDNA, 20 µM nt. (C) Purified Rad51 (WT, E206A, or EED) was incubated with purified Swi5-Sfr1 (or the equivalent volume of protein storage buffer) and subjected to immunoprecipitation with an anti-Sfr1 antibody. For quantification, Rad51 signal was normalized to Sfr1 signal and expressed relative to wild type. (D) Strand exchange reactions were conducted according to the scheme outlined above the gel. Rad51 (WT, E206A, or EED), 5 µM. Swi5-Sfr1 (S5S1), indicated. RPA, 1 µM. cssDNA, 10 µM nt. ldsDNA, 10 µM nt. Data in (B,D) are means of three independent experiments and error bars depict standard deviation. Data in (C) are the means of two independent experiments with individual values shown.

The online version of this article includes the following source data and figure supplement(s) for figure 6:

**Source data 1.** Strand exchange yield (%) in *Figure 6B,D* and relative co-IP of Rad51 (%) for *Figure 6C*.

**Figure supplement 1.** Rad51-EED binds both ssDNA and dsDNA like wild-type Rad51.

**Figure supplement 1—source data 1.** Relative co-IP of Rad51 (%) for *Figure 6—figure supplement 1D*.

immunoprecipitate, indicating that the EED mutation ablates Rad51-Rad52 complex formation in vivo. These results suggest that the exquisite DNA damage sensitivity of *rad51-EED* is due to defects in the interactions with both Rad55-Rad57 and Rad52. The identity of the band indicated as Rad52 was verified by using a *rad52Δ* strain (*Figure 7—figure supplement 1A*). To directly test whether the EED mutation disrupts the binding of Rad51 to Rad52, a co-IP experiment was conducted with purified proteins. Comparable amounts of Rad52 co-IP'd with Rad51 and Rad51-E206A (*Figure 7B*).

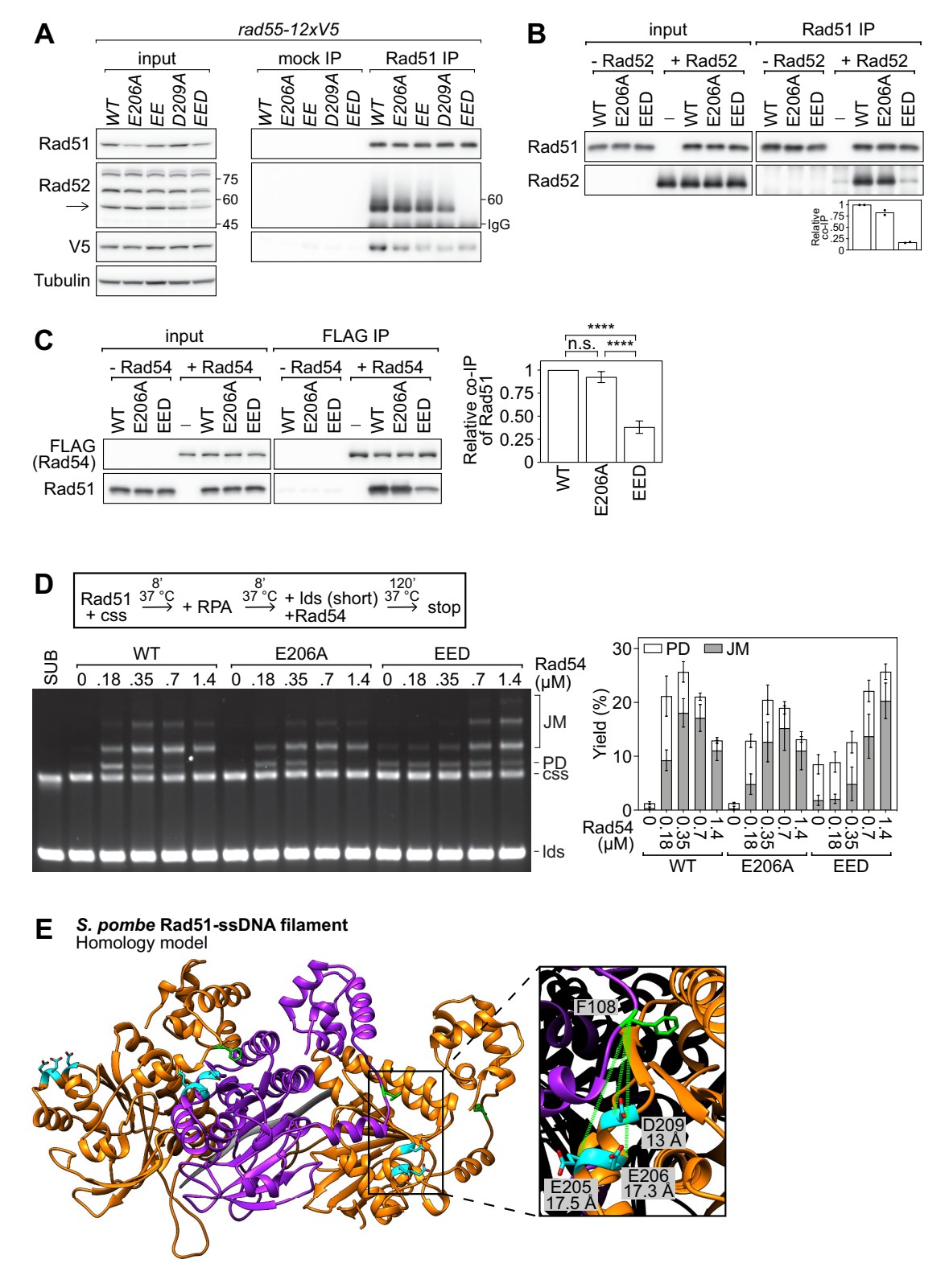

**Figure 7.** The PAP facilitates the interaction of Rad51 with Rad52 and Rad54. (**A**) Soluble cell extracts treated with a benzonase-like nuclease were prepared from each strain under native conditions (input). Immunoprecipitation (IP) was then performed with mock (human IgG from non-immunized animal) or anti-Rad51 antibodies. Arrow denotes the Rad52 band (other bands were still observed in a *rad52Δ* strain, see *Figure 7—figure supplement 1A*) and numbers denote the position of the size markers in kDa. To ensure separation of Rad52 from the IgG heavy chain band, the IP samples were

*Figure 7 continued on next page*

*Figure 7 continued*

separated by 7% SDS-PAGE instead of the 12% SDS-PAGE employed for the input samples. As a result, the 45 kDa and 75 kDa bands of the size marker are not within the cropped area of the anti-Rad52 immunoblot for the IP sample. Tubulin serves as a loading control. (B) Purified Rad51 (WT, E206A, or EED) was incubated with purified Rad52 (or the equivalent volume of protein storage buffer) and subjected to immunoprecipitation with anti-Rad51 antibody. For quantification, Rad51 signal was normalized to Sfr1 signal and expressed relative to wild type. (C) Purified Rad51 (WT, E206A, or EED) was incubated with purified FLAG-Rad54 (or the equivalent volume of protein storage buffer) and subjected to immunoprecipitation with FLAG-agarose resin. For quantification, Rad51 signal was normalized to FLAG-Rad54 signal and expressed relative to wild type. Statistical significance was assessed by one-way ANOVA with Tukey's multiple comparisons test. n.s., not significant (p=0.2538). ****p<0.0001. (D) Strand exchange reactions were conducted according to the scheme outlined above the gel. Rad51 (WT, E206A, or EED), 7 μM. Rad54, indicated. RPA, 0.7 μM. cssDNA, 7 μM nt. ldsDNA, 14 μM nt. (E) Ribbon depiction of three *Sp*Rad51 monomers (alternating orange and purple) bound to ssDNA (9-mer poly-dT in gray). Residues E205, E206, and D209 are colored in cyan with their side-chains shown (O atoms in red), and F108 is colored in green with its side-chain shown. The α-helix of the rightmost monomer containing E205, E206, and D209 is enlarged, along with the F108 residue of the central monomer, to illustrate the close proximity of these residues at the subunit interface. Dotted-green lines show the $C_\alpha$-$C_\alpha$ distance between each residue and F108. The Rad51 filament is employed as a proxy to demonstrate that insertion of Phe in the FxxA motif of auxiliary factors could be facilitated by interactions with the PAP. Data in (B) are means of two independent experiments with individual values shown. Data in (C,D) are means of three independent experiments and error bars depict standard deviation.

The online version of this article includes the following source data and figure supplement(s) for figure 7:

**Source data 1.** Relative co-IP of Rad52 and Rad51 in *Figure 7B,C*, and strand exchange yield (%) in *Figure 7D*.
**Figure supplement 1.** Anti-Rad52 antibody verification and Rad54 purity analysis.
**Figure supplement 2.** The PAP is conserved in eukaryotic Dmc1 and archaeal RadA.
**Figure supplement 3.** The PAP is not conserved in bacterial RecA.

By stark contrast, only background levels of Rad52 were seen to co-IP with Rad51-EED. Similar results were obtained in the reciprocal co-IP experiment (*Figure 7—figure supplement 1B*). These results clearly indicate that the PAP is essential for the Rad51-Rad52 interaction.

Our results demonstrated that the PAP is involved in the interaction of Rad51 with both Rad55-Rad57 and Rad52. We therefore considered the possibility that the PAP is commonly utilized by multiple auxiliary factors and sought to examine the interaction of Rad51 with Rad54. Unlike Rad55-Rad57 and Rad52, Rad54 functions primarily in the later stages of HR (*Sugawara et al., 2003*; *Lisby et al., 2004*). Defects in the recruitment of Rad51 to DNA damage sites—as seen in *rad51-EED*—would block downstream events, obscuring analysis of the Rad51-Rad54 interaction. To circumvent this, Rad54 with an N-terminal FLAG tag was purified to near-homogeneity from *E. coli* (*Figure 7—figure supplement 1C*) and its physical interaction with Rad51-EED was examined in vitro. Similar amounts of Rad51 and Rad51-E206A were seen to co-IP with Rad54, whereas an approximately threefold reduction in the co-IP of Rad51-EED was observed (*Figure 7C*). To examine whether this difference is of functional significance, strand exchange reactions were conducted under conditions where wild-type Rad51 alone did not produce JM or PD molecules (Rad51-EED nevertheless showed increased intrinsic strand exchange activity [*Figure 7D*], as expected from our previous experiments [*Figure 6A,B*]). Remarkably, the EED mutation rendered Rad51 less sensitive to the stimulatory effect of Rad54, with approximately fourfold more Rad54 required to achieve maximal activity. These results indicate that, in addition to facilitating the interaction of Rad51 with Rad55-Rad57 and Rad52, the PAP is also important for the interaction with Rad54.

## Discussion

To understand more about the regulation of Rad51, we employed a structural approach to identify regions of Rad51 that are important for its interaction with auxiliary factors. This highlighted a protruding acidic patch, which we refer to as the PAP, comprised of residues E205, E206, and D209 on the exterior of the Rad51 filament (*Figure 1*). Genetic analysis strongly suggested that mutation of the most conserved residue in this patch (*rad51-E206A*) specifically impairs the interaction between Rad51 and Rad55-Rad57 without affecting the interaction with Swi5-Sfr1 (*Figure 2*), and this was bolstered by biochemical reconstitutions (*Figure 3*). Strikingly, mutation of all three PAP residues to Ala (*rad51-EED*) completely abolished recombinational DNA repair, leading to a phenotype more severe than *rad57Δ* (*Figure 4*). Mechanistically, the DNA damage sensitivity of *rad51-EED* was found to stem from defects in the recruitment of Rad51 to DNA damage sites (*Figure 5*). Biochemical reconstitutions suggested that Rad51-EED is not intrinsically defective and retains a functional

interaction with Swi5-Sfr1 (*Figure 6*). Remarkably, the EED mutation was found to abrogate the interaction of Rad51 with Rad52 and impair the interaction with Rad54 (*Figure 7*). Our results indicate that the PAP, a novel motif of Rad51, is critically important for the interaction with Rad55-Rad57, Rad52, and Rad54. We propose that the PAP is a fundamental motif commonly utilized by recombination auxiliary factors to potentiate Rad51.

## The PAP of Rad51 is an evolutionarily conserved motif that is integral to recombinational DNA repair

E205, E206, and D209 were found to form the PAP on the exterior of the Rad51-ssDNA filament (*Figure 1A–C*). Individual mutations within the PAP impaired DNA repair to differing extents. D209A had the highest penetrance, whereas E206A, and to a much lesser extent E205A, conferred sensitivity only in the *sfr1Δ* background, where DNA repair is strictly dependent on Rad55-Rad57 (*Figure 4A and C* and *Figure 4—figure supplement 1C*). These genetic data are consistent with the notion that the PAP is more relevant to the function of Rad55-Rad57 than Swi5-Sfr1. When both Glu residues were mutated to Ala (*rad51-EE*), moderate DNA damage sensitivity was observed even in the presence of both Rad55-Rad57 and Swi5-Sfr1; this synergism implies that the functions of E205 and E206 are related, and that the ability of Sfr1 to suppress the DNA damage sensitivity conferred by E206A is partially dependent on E205. Moreover, mutating all three residues to Ala (*rad51-EED*) completely abolished Rad51-dependent DNA repair. The additive effect of combining mutations suggests that the overall negativity of the PAP is important for its function. This is bolstered by the finding that the replacement of E206 with a positively charged Lys residue is highly disruptive, phenocopying the neutralization of all three residues via Ala mutations (*Figure 4—figure supplement 1D*). Thus, while all three residues are relevant to PAP function, there appears to be a hierarchy: D209A is more important than E206, which itself is more important than E205. This suggests that the closer a residue is to the C-terminal end of the α-helix (top of the helix in our structural models), the more important it is for PAP function (discussed below).

Negatively charged regions were seen in the equivalent α-helices of both *Sc*Rad51 and *Hs*Rad51 (*Figure 1—figure supplement 1B,C*), indicating structural conservation of the PAP. Furthermore, sequence alignments suggest that the PAP is widely conserved among eukaryotes, with residues E206 and D209 each showing conservation in seven-out-of-eight eukaryotic Rad51 orthologues (*Figure 7—figure supplement 2A*). Interestingly, examination of a structural model revealed conservation of the PAP in Dmc1, the meiosis-specific RecA-family recombinase (*Figure 7—figure supplement 2B*). The archaeal RadA polymer also exhibited substantial structural conservation of the PAP (*Figure 7—figure supplement 2C*). By contrast, this region showed poor sequence conservation in bacterial RecA, and although a short α-helix at the equivalent position appeared to exist in the RecA monomer (*Figure 7—figure supplement 3A*), a loop was seen instead in the RecA-ssDNA filament structure (*Figure 7—figure supplement 3B*). Furthermore, despite this region being negatively charged, this was partially due to the close proximity of residues E32 and D33 from the adjacent RecA monomer. We interpret these analyses to mean that the PAP is conserved in eukaryotic Rad51/Dmc1 and archaeal RadA, but not in bacterial RecA. While the mediator and ssDNA annealing activities of Rad52 are provided by RecFOR in bacteria, only eukaryotes and archaea also possess recombinase paralogs and Rad54 (*Zelensky et al., 2014*). We postulate that the PAP may have evolved in organisms where the RecA-family recombinase is potentiated by multiple distinct auxiliary factors. Consistently, it is notable that Dmc1 is subjected to extensive regulation by auxiliary factors: Swi5-Sfr1, Hop2-Mnd1, and Rdh54 simulate Dmc1 (*Haruta et al., 2006*; *Chi et al., 2009*; *Tsubouchi et al., 2020*); and Rad52 has been proposed to inhibit Dmc1 (*Murayama et al., 2013*).

## The PAP plays a central role in the interaction of Rad51 with recombination auxiliary factors

The PAP-containing α-helix precedes a β-strand that is critically important for Rad51 polymerization (*Pellegrini et al., 2002*; *Shin et al., 2003*; *Conway et al., 2004*). This β-strand is involved in facilitating interactions with the FxxA consensus sequence within the inter-domain linker of the adjacent monomer (corresponding to $^{108}$FTTA$^{111}$ in *Sp*Rad51). The Phe residue of the adjacent monomer inserts into a hydrophobic pocket above the β-strand, which stabilizes inter-subunit contacts. Because some auxiliary factors that contain the FxxA motif employ Rad51 mimicry to interact with

Rad51, FxxA has been proposed to function as a Rad51 interaction motif (*Pellegrini et al., 2002*; *Shin et al., 2003*). Neither Swi5 nor Sfr1 contain the FxxA consensus sequence and we recently identified two non-FxxA sites within the intrinsically disordered N-terminal half of Sfr1 that are responsible for the binding of Swi5-Sfr1 to Rad51 (*Argunhan et al., 2020*). By contrast, both Rad57 ($^{145}$FELA$^{148}$) and Rad52 ($^{17}$FNTA$^{20}$ and $^{338}$FISA$^{341}$) contain the FxxA consensus sequence, as does Rad54 ($^{804}$FIRA$^{807}$). Rad55 does not contain an FxxA motif, and notably, Y2H analysis suggested that Rad51 interacts with Rad57 but not Rad55 (*Tsutsui et al., 2001*). Furthermore, Y2H analysis mapped the minimal *Sp*Rad51-interacting domain of Rad52 to a C-terminal fragment (310-469) containing the $^{338}$FISA$^{341}$ sequence (*Kim et al., 2002*), and mutation of the corresponding Phe residue to Ala in *Sc*Rad52 (F349A) completely abrogated its binding to Rad51 (*Kagawa et al., 2014*). Taken together, these observations strongly suggest that Rad52 employs the FxxA motif to bind Rad51.

Our results demonstrate that neutralization of the PAP impairs the interaction with Rad55-Rad57, Rad52, and Rad54, while the interaction with Swi5-Sfr1 is mostly unaffected (*Figures 4B*, *6C, D* and *7A–D* and *Figure 7—figure supplement 1B*). We cannot exclude the possibility that the PAP has some role in facilitating the interaction with Swi5-Sfr1, but a simple inference that can be drawn from these results is that the PAP is particularly important for auxiliary factors that employ Rad51 mimicry to modulate Rad51. The close proximity of the PAP to the Phe residue within the FxxA motif of the adjacent monomer—$C_\alpha$-$C_\alpha$ distances for PAP residues and *Sp*Rad51-F108 are predicted to be 13.0 Å (D209), 17.3 Å (E206), and 17.5 Å (E205)—means it would be well-placed to influence interactions with auxiliary factors that employ Rad51 mimicry (*Figure 7E*). We speculate that this may be the mechanism through which the PAP exerts its effects on Rad51 potentiation. Since the negative charge of the PAP is important for recombinational DNA repair (*Figure 4A* and *Figure 4—figure supplement 1D*), it is likely that the PAP participates in electrostatic interactions that are important for the modulation of Rad51 by these auxiliary factors, which may utilize the PAP as a landing pad to facilitate FxxA insertion. We anticipate that a basic patch may exist that is close in three-dimensional space to the FxxA motif of each auxiliary factor, and this basic patch may be critically important for interacting with the PAP and facilitating the binding to Rad51. Interestingly, Y2H analysis by *Kim et al., 2002* isolated several other mutations in *Sp*Rad51 (G177S, C179F, G282D, and L274P) that disrupt the interaction with Rad52, and these residues are also in reasonably close proximity to F108 (*Figure 7—figure supplement 3C*). Notably, the G177S and C179F mutations also impaired the interaction with Rad54. Moreover, the C179F mutation even abrogated the interaction with Rad57 and the self-association of Rad51. Given the close proximity of G177 and C179 to the PAP ($C_\alpha$-$C_\alpha$9–12 Å), it is possible that mutation of these residues affects PAP function and/or F108 insertion. These Y2H results are consistent with our proposal that multiple auxiliary factors interact with Rad51 via a mechanism involving the PAP.

PAP mutations impaired complex formation with Rad55-Rad57 in vivo (*Figures 2E*, *4B* and *7A*), but even in *rad51-EED*, complex formation was not completely abolished. By contrast, while formation of the Rad51-Rad52 complex was mostly unaffected by mutation of only the Glu residues within the PAP, the D209A mutation led to a marked reduction in complex formation (*Figure 7A*). Consistent with the notion that the PAP as a whole is important, the EED mutation completely abrogated Rad51-Rad52 complex formation. Co-IP experiments with purified proteins directly demonstrated that the interaction of Rad51 with both Rad52 and Rad54 is facilitated by the PAP (*Figure 7B–D* and *Figure 7—figure supplement 1B*). Performing such experiments with Rad55-Rad57 awaits the purification of this biochemically intractable complex.

If multiple auxiliary factors utilize the same motif to potentiate Rad51, they would be expected to compete for Rad51 binding. However, because auxiliary factors perform non-overlapping roles in HR, the requirement for binding Rad51 is likely to be temporally distinct, with Rad52 being the most upstream binding partner, followed by Rad55-Rad57, then Rad54 (*Sugawara et al., 2003*; *Lisby et al., 2004*). Furthermore, the Rad51 nucleoprotein filament contains as many PAP motifs as there are monomers in the filament, providing ample opportunities for auxiliary factors to bind Rad51 in a non-competitive manner. Notably, the PAP is important for interactions with auxiliary factors involved in both the early (Rad52 and Rad55-Rad57) and late (Rad54) stages of DNA strand exchange, indicating that the PAP is integral to the role of Rad51 in HR. Further research is needed to examine the temporospatial coordination of Rad51 binding, especially in light of the finding that some auxiliary factors such as Rad55-Rad57 and Swi5-Sfr1 interact with each other (*Argunhan et al., 2020*).

### *rad51-E206A* uncovers insights into the interplay between Rad55-Rad57 and Swi5-Sfr1

Substantial genetic analysis suggests that the defects of *rad51-E206A* stem specifically from an impairment in the interaction with Rad55-Rad57. This is supported by biochemical analysis demonstrating that the intrinsic strand exchange activity of Rad51-E206A is comparable to wild type, and its interaction with Swi5-Sfr1, Rad52, and Rad54 is not grossly impaired. Interestingly, the *rad51-E206A* mutant was proficient for DNA repair in the presence of Sfr1, but completely defective in its absence (*Figure 2A–C* and *Figure 2—figure supplement 1A–C*). This suppression did not occur through enhancing Rad51–Rad55-Rad57 complex formation (*Figure 2E*), suggesting that Swi5-Sfr1 can functionally compensate for defects in the physical association of Rad51 with Rad55-Rad57. Swi5-Sfr1 may function in a contingency capacity for Rad55-Rad57 such that a moderate reduction in Rad51–Rad55-Rad57 complex formation does not affect DNA repair as long as Swi5-Sfr1 is present. Suppression of the DNA damage sensitivity associated with *rad51-E206A* by Sfr1 contradicts the commonly accepted model wherein Rad55-Rad57 and Swi5-Sfr1 comprise independent sub-pathways of HR (*Akamatsu et al., 2003*; *Akamatsu et al., 2007*). We recently showed that Rad55-Rad57 can suppress defects in the interaction between Swi5-Sfr1 and Rad51, and that Swi5-Sfr1 physically interacts with Rad55-Rad57 (*Argunhan et al., 2020*). Thus, there is increasing evidence to suggest that Rad55-Rad57 and Swi5-Sfr1, while capable of functioning independently of each other, collaboratively promote Rad51-dependent DNA repair.

In summation, we have characterized an acidic patch of Rad51 that comprises an evolutionarily conserved motif important for the interaction of Rad51 with its major auxiliary factors: the Rad51 paralogs Rad55-Rad57, Rad52, and Rad54. While this motif is essential for HR in *S. pombe*, the extent to which it is required in other organisms remains to be determined and will likely be a focal point of future research.

## Materials and methods

### Sequence alignments and structural models

Sequence alignments were prepared using Clustal Omega (UniProt identifiers are: *S. pombe*, P36601; *S. cerevisiae*, P25454; *U. maydis*, Q99133; *C. elegans*, G5EGG8; *D. melanogaster*, Q27297; *G. gallus*, P37383; *M. musculus*, Q08297; *H. sapiens*, Q06609). All structural depictions were prepared using UCSF Chimera (*Pettersen et al., 2004*). The Coulombic Surface Coloring feature in UCSF Chimera was employed to depict surface charge. The model of an *Sp*Rad51 monomer was generated by Phyre2 (intensive mode; *Kelley et al., 2015*) based on two known structures of *Sc*Rad51 (*Conway et al., 2004*; *Chen et al., 2010*), with Protein databank (PDB) identifiers 1SZP and 3LDA; and one known structure of *Hs*Rad51 (*Short et al., 2016*) with PDB identifier 5JZC. 46 residues (1–41 and 361–365) of the *Sp*Rad51 monomer were modeled ab initio and these low confidence regions were omitted from the model. The *Sp*Rad51-ssDNA filament model consisting of three monomers is a homology model that was described previously by *Ito et al., 2020*. This structure was previously deposited in the Biological Structure Model Archive (identification code BSM00017). The structure of Rad51-EED was analyzed by substituting E205, E206, and D209 to Ala through the Rotamer feature. The model of an *Sp*Dmc1 monomer was also generated by Phyre2 (intensive mode; *Kelley et al., 2015*) based on three known structures of RadA (*Shin et al., 2003*; *Wu et al., 2004*; *Chen et al., 2007*), with PDB identifiers 1PZN, 1T4G, and 2DFL; two known structures of *Sc*Rad51 (*Conway et al., 2004*; *Chen et al., 2010*), with PDB identifiers 1SZP and 3LDA; and one known structure of *Hs*Rad51 (*Short et al., 2016*) with PDB identifier 5JZC. Residues 1–14 were modeled ab initio and were therefore omitted from the model.

The *Hs*Rad51-ssDNA filament shown is the structure resolved by cryo-electron microscopy (*Xu et al., 2017*; PDB 5H1B), whereas the *Sc*Rad51-ssDNA filament shown is the structure solved by X-ray crystallography (*Conway et al., 2004*; PDB 1SZP, chains E and F). Note that ssDNA density is missing from the structure of the *Sc*Rad51-ssDNA filament model, despite the formation of crystals in the presence of ssDNA (*Conway et al., 2004*). The structure of the *Pyrococcus furiosus* RadA polymer was solved by X-ray crystallography (*Shin et al., 2003*; PDB 1PZN, chains G, A, and B). The RecA monomer (*Story et al., 1992*; PDB 2REB) and ssDNA filament (*Chen et al., 2008*; 3CMU) structures were solved by X-ray crystallography.

### *Schizosaccharomyces pombe* strains

*S. pombe* strains used in this study are listed in the Key Resources Table. All strains are isogenic derivatives of strain YA119 (*Akamatsu et al., 2003*). For most assays, strains of the h minus mating type were employed (*Msmt-0*), except in *Figure 1D* (*mat1PD17::LEU2*). Standard media was used for growth (YES), selection (YES with drugs or EMM), and sporulation (SPA), as described previously (*Hentges et al., 2005*). All reasonable requests for strains will be fulfilled by the co-corresponding authors (B. Argunhan and H. Iwasaki).

Primers used in this study are listed in the Key Resources Table. In order to introduce the *rad51-E206A* mutation at the native *rad51+* locus, plasmid p6 (*pET11b-rad51+*; *Haruta et al., 2006*) was amplified with mutagenic primers 398–400 and 399–405 to produce amplicons 1 and 2, respectively. Amplicon 3, containing the *ADH1* terminator ($T_{ADH1}$) and *kanMX6* cassette, was amplified from the C-terminal epitope tagging plasmid p51 (*pFA6a-13xMYC-kanMX6*) using primers 350–351. Regions upstream and downstream of the *rad51+* locus were amplified with primers 409–412 and 417–419 to yield amplicons 4 and 5, respectively. All five amplicons were cloned onto the pBlueScript II SK (+) vector using the In-Fusion Cloning Kit (Takara) to generate plasmid pNA46. pNA46 was then cut with restriction enzymes MscI and XhoI (NEB) and the fragment containing *rad51-E206A-$T_{ADH1}$-kanMX6* was gel extracted and transformed into strain NA310 (*wild type*). *rad51-E205A, rad51-EE, rad51-D209A, rad51-EED,* and *rad51-E206K* were made in exactly the same way using plasmids pBA144, pBA146, pBA145, pBA148, and pBA152, respectively. To generate the *rad55-12xV5* strain, plasmid p85 (*pNX3c-PK12*; *Amelina et al., 2016*) was PCR-amplified with primers 200–201 and the wild-type strain (NA310) was transformed with this amplicon. All cloning and genetic manipulations were confirmed by DNA sequencing. Sequence alignments were prepared with Clustal Omega.

### DNA damage sensitivity assays

A single colony was streaked as a patch onto a YES plate and grown at 30℃ for 1–2 days. This patch was then resuspended in 2 mL of YES and grown with shaking at 30℃ for ~24 hr. Cells were seeded into 2 mL of fresh YES ($0.25 \times 10^6$ cells/mL for *rad+*; $0.5 \times 10^6$ cells/mL for *rad-*) and grown with shaking at 30℃ until they reached log phase. For spot tests, cell density was adjusted to $2 \times 10^7$ cells/mL and tenfold serial dilutions were made. 5 µL of each dilution was spotted onto YES plates without drugs and YES plates containing DNA damaging agents. For UV treatment, a YES plate without drugs was UV-irradiated. Cells were incubated at 30℃ or 21℃, as indicated. For clonogenic survival assays, cells were prepared as described above and spread onto YES plates, which were then exposed to acute UV irradiation of the specified dose. Colonies were counted after 3–4 days of incubation at 30℃ and survival percentage was expressed relative to the number of colonies on the untreated plate. Statistical analysis was by unpaired two-tailed t-test using GraphPad Prism (version 8).

### Extraction and detection of cellular proteins

A single colony was streaked as a patch on a YES plate and grown for 1–2 days at 30℃. This patch was resuspended in 2 mL of YES and grown with shaking at 30℃ for 24 hr. Cells were seeded into 70 mL of fresh YES ($0.15 \times 10^6$ cells/mL for *rad+*; $0.30 \times 10^6$ cells/mL for *rad-*) and grown in 500 mL baffled flasks with shaking at 30℃ until they reached log phase. Cells were harvested and cell pellets were resuspended in 35 mL of sterile water and divided into two equal aliquots (±UV). +UV aliquots were transferred into a petri dish at a depth of ~0.5 cm and exposed to UV ($200 \text{ J/m}^2$), while -UV aliquots were mock treated. Each aliquot was then resuspended in 35 mL fresh YES and allowed to recover at 30℃ for 3 hr with shaking. Cells were then harvested and treated as previously described (*Argunhan et al., 2020*). Briefly, $1 \times 10^8$ cells were resuspended in 1 mL of ice-cold water and mixed with 150 µL of 1.85 M NaOH 7.5% β-mercaptoethanol on ice for 15 min. 150 µL of 55% TCA was then added, followed by mixing and a further 10 min incubation on ice. Precipitated proteins were pelleted by sequential centrifugation (20000 *g*, 10 min, 2℃; then 20000 *g*, 1 min, 2℃). Pellets were resuspended in 100 µL of urea buffer (8 M urea, 5% SDS, 200 mM Tris-Cl pH 6.8, 1 mM EDTA, 0.01% BPB) freshly supplemented with 0.1 M DTT and 0.2 M Tris and dissolved on a thermomixer (65℃ 10 min 1300 RPM). Proteins were then separated by SDS-PAGE and transferred to PVDF membranes. Antibodies against Rad51 (1:10,000; *Argunhan et al., 2020*) and Tubulin (1:10,000; Sigma T5168) were employed. Horseradish peroxidase (HRP)-conjugated secondary antibodies were

purchased from GE Healthcare (mouse, 1:5,000, NA931; rabbit, 1:5,000, NA934) or Jackson Immunoresearch Labs (rat, 1:10000, 712-035-153).

## In vivo immunoprecipitation (IP)

A single colony was resuspended in 20 mL YES and grown with shaking at 30℃ (24 hr for $rad^+$; 48 hr for $rad^-$). Cells were seeded into 1 L of fresh YES ($0.2 \times 10^6$ cells/mL for $rad^+$; $0.4 \times 10^6$ cells/mL for $rad^-$) in 5 L baffled flasks and grown with shaking at 30℃ until they reached log phase. Cultures were then supplemented with 0.5 mM PMSF and harvested by centrifugation (6000 $g$, 15 min, 2℃), and pelleted cells were washed with 20 mL of yeast wash solution (50 mM HEPES [pH 7.5], 70 mM KOAc, 1 mM PMSF). Aliquots of $2 \times 10^9$ cells were pelleted, frozen in liquid nitrogen, and stored at −80℃ until use. IP experiments were then carried out as previously described (*Argunhan et al., 2017b*). Briefly, a single cell pellet was resuspended in 400 µL of KA50 buffer (50 mM HEPES-KOH [pH 7.5], 50 mM KOAc, 5 mM MgOAc, 0.05% Igepal CA-630, 10% glycerol, 0.25 mM TCEP, 0.5 mM PMSF, 2x cOmplete protease inhibitor cocktail [Roche], 10 mM β-glycerophosphate, 10 mM NaF, 1 mM sodium orthovanadate) and mixed with an equal volume of glass beads (500 micron) on ice. Cells were then broken using a Yasui Kikai Multi-beads Shocker (2700 RPM, 30 s on/off, 12 cycles, 2℃). The lysate was recovered, and the glass beads were washed with 200 µL of KA50 buffer and combined with the lysate. The lysate was then treated with 250 units of TurboNuclease (Accelagen) for 30 min at 4℃ with mixing. The lysate was then sequentially cleared (20,000 g, 10 min, 2℃; then 20,000 g, 5 min, 2℃), a sample was taken for immunoblotting (input) and mixed with an equal volume of 2x SDS loading buffer (120 mM Tris-HCl [pH 6.8], 4% SDS, 20% glycerol, 0.02% BPB, 200 mM DTT), and the remaining soluble cell extract was divided into three equal aliquots in protein Lo-Bind tubes (Eppendorf). Anti-V5 (mouse; MCA1360 Bio-Rad), anti-Rad51 (rabbit; *Haruta et al., 2006*), or ChromPure Human IgG (mock antibody; 009-000-003 Jackson Immunoresearch Laboratories) antibodies were premixed with Dynabeads Protein A (Thermo Fisher Scientific) and incubated with each aliquot of soluble cell extract with gentle mixing (3 hr, 4℃). Aqueous fractions were separated using a magnetic stand and the beads were briefly washed with buffer KA50 (300 µL, x3). The beads were then resuspended in 75 µL of 1x SDS loading buffer and proteins were eluted using a thermomixer (IP; 65℃ 10 min 1300 RPM). Samples were separated by SDS-PAGE, transferred to PVDF membranes and detected with the following antibodies: anti-V5, mouse (1:10,000; Bio-Rad); anti-Rad51, rat (1:10,000; *Argunhan et al., 2020*); anti-Rad52, rabbit (1:5,000; *Kurokawa et al., 2008*); and anti-Tubulin, mouse (1:10,000, T5168 Sigma-Aldrich). HRP-conjugated secondary antibodies were purchased from GE Healthcare (mouse, 1:5000, NA931; rabbit, 1:5,000, NA934) or Jackson Immunoresearch Labs (rat, 1:10,000, 712-035-153). FIJI software (*Schindelin et al., 2012*) was used for quantification of in vivo IPs. Briefly, background subtraction was performed by the rolling ball method and the signal for IP'd and co-IP'd protein was quantified. The co-IP'd signal was then divided by the IP'd signal and expressed relative to wild type. Graphs were prepared using GraphPad Prism (version 8).

## Protein purification

*E. coli* BL21 (DE3) RIPL strain was used for purification of *S. pombe* Rad51, RPA, Swi5-Sfr1, and Rad52. Rad51 (plasmid p6; *pET11b-rad51+*), Rad51-E206A (plasmid pNA42; *pET11b-rad51-E206A*), and Rad51-EED (plasmid pBA161; *pET11b-rad51-E205A-E206A-D209A*) were expressed with 1 mM IPTG at 18℃ for ~14 hr and purified exactly as previously described (*Kurokawa et al., 2008*). Rad51 mutants behaved similarly to wild-type Rad51 throughout the purification process. *S. pombe* RPA was expressed from plasmid p69 (*pET11b-ssb2-ssb3-ssb1*) and Swi5-Sfr1 was expressed from plasmid p71 (*pBKN220-sfr1-swi5*) as for Rad51, and purified exactly as previously described (*Haruta et al., 2006*; *Kurokawa et al., 2008*). Rad52 was expressed from plasmid p76 (*pET11b-rad52+*) with 1 mM IPTG at 30℃ for 3 hr and purified exactly as previously described (*Kurokawa et al., 2008*).

*S. pombe* Rad54 fused to an N-terminal tag—containing a hexahistidine tag, the Fh8 fusion partner (*Costa et al., 2013*), and a FLAG epitope, with a PreScission protease recognition site between the Fh8 and FLAG components (pBA110; *pET15b-6xHis-Fh8-PreScission-1xFLAG-rad54+*)—was expressed in 20 L of *E. coli* strain BL21 (DE3) Star at an OD of ~0.45 with 1 mM IPTG at 18℃ for ~14 hr. Cells were harvested by centrifugation, washed with *E. coli* wash solution (50 mM Tris-Cl [pH 7.5],

150 mM NaCl, 1 mM PMSF), and stored at −80°C until required. Cell pellets (~53 g) were then resuspended in 250 mL of R buffer (20 mM Tris-Cl [pH 7.5], 10% glycerol, 1 mM EDTA) containing 500 mM NaCl, 5 mM MgOAc, 0.5 mM ATP, 2 mM imidazole, 1 mM DTT, 0.5 mM PMSF, and 0.01% igepal CA-630. Cells were then disrupted by sonication and the lysate was cleared by ultracentrifugation (70,000 g 1 hr 2°C). The clarified lysate was incubated with 10 mL of cOmplete His-Tag Purification Resin (Roche) with gentle mixing (1 hr 4°C). The resin was poured into a glass column (Bio-Rad Econo-column, 2.5 cm x 10 cm) and washed sequentially with the same buffer used for lysis (50 mL x4) and R buffer containing 500 mM NaCl, 5 mM imidazole, 1 mM DTT, and 0.5 mM PMSF (50 mL x8). Proteins were then eluted in R buffer containing 300 mM NaCl, 300 mM imidazole, and 0.5 mM TCEP (2 mL x4). Eluates were combined, supplemented with PreScission protease (GE Healthcare), and dialyzed against 1 L of the same buffer without imidazole (overnight 4°C). The protein sample was diluted in 2x volumes of R buffer containing 0.5 mM TCEP and applied to a 1 mL HiTrap Q column equilibrated with R buffer containing 100 mM NaCl and 0.5 mM TCEP. Rad54, which was found in the flow-through, was then applied to a 1 mL HiTrap Heparin column equilibrated with R buffer containing 100 mM NaCl and 0.5 mM TCEP. Proteins were then eluted with a linear gradient (20 mL 0.1–0.6 M NaCl). Peak fractions containing Rad54 were combined, diluted with 9x volumes of R buffer containing 0.5 mM TCEP, then applied to a 1 mL Resource S column equilibrated with R buffer containing 50 mM NaCl and 0.5 mM TCEP. Proteins were eluted with a linear gradient (20 mL 0.05–0.7M NaCl), then peak fractions were pooled and developed in a 16/60 Superdex 200 PG gel filtration column in R buffer containing 300 mM NaCl and 0.5 mM TCEP. Peak fractions were pooled, diluted with 2x volumes of R buffer containing 0.5 mM TCEP, and applied to a 1 mL Resource S column. Proteins were eluted with a linear gradient (40 mL 0.1–0.5 M NaCl). Rad54 eluted at ~220 mM NaCl and was subsequently concentrated using a Vivaspin six centrifugal concentrator (MWCO 30 kDa). The concentration was estimated by measuring the A280 with a molar extinction coefficient of 72530 $M^{-1}$ $cm^{-1}$. The concentrated protein was frozen in small aliquots using liquid nitrogen and stored at −80°C until required. The yield of highly purified Rad54 was ~0.5 mg. Chromatography columns were purchased from GE Healthcare.

## Electrophoretic mobility shift assay (EMSA)

Rad51 (wild type, Rad51-E206A, or Rad51-EED) was incubated with 30 micromolar nucleotide (μM nt) of PhiX174 virion DNA or 20 μM nt of ApaLI-linearized PhiX RFI DNA in EMSA buffer (30 mM HEPES-KOH [pH 7.5], 150 mM KCl, 3 mM MgCl$_2$, 2 mM ATP, 1 mM DTT, 5% glycerol). The 10 μL reaction was incubated at 37°C for 15 min. 1.1 μL of 2% glutaraldehyde was then added and incubation was continued for a further 5 min at 37°C. 2.5 μL of loading dye was added and 3 μL of the reaction was loaded onto a 0.8% agarose gel in TAE buffer (50 V 120 min). The gel was then stained with SYBR Gold (Thermo Fisher Scientific) and imaged using a LAS4000 mini (GE Healthcare).

## ATPase assay

The ATPase assay was conducted exactly as previously described (*Argunhan et al., 2020*). Briefly, 5 μM of Rad51 or Rad51-E206A was mixed with 10 μM nt of PhiX174 virion DNA on ice in ATPase buffer (30 mM Tris-Cl [pH 7.5], 100 mM KCl, 3.5 mM MgCl$_2$, 1 mM DTT, 5% glycerol), and either 0.5 μM Swi5-Sfr1 or the equivalent volume of protein storage buffer was added. Reactions were then initiated through the addition of 0.5 mM ATP and 10 μL aliquots were withdrawn at multiple timepoints spanning the initial reaction rate, immediately mixed with 2 μL of 120 mM EDTA and stored at room temperature. The concentration of inorganic phosphate was then measured with a commercial malachite green phosphate detection kit (BioAssay Systems) according to the manufacturer's instructions.

## Three-strand exchange assay

To examine the intrinsic strand exchange activity of Rad51, 15 μM of Rad51 (wild type, Rad51-E206A, or Rad51-EED) was incubated with 30 μM nt PhiX174 virion DNA (NEB) for 5 min at 37°C in strand exchange buffer (30 mM Tris-HCl [pH7.5], 1 mM DTT, 150 mM KCl, 3.5 mM MgCl$_2$, 2 mM ATP, 8 mM phosphocreatine, 8 units/mL creatine phosphokinase and 2.5% glycerol). Next, 1 μM of RPA was added to the reaction and after a 5 min incubation, the reaction was initiated with 20 μM nt of PhiX RFI DNA (NEB) linearized with ApaLI. In the case of the shortened lds substrate, a 1.6 kb

dsDNA fragment was employed instead, and this was prepared by digesting PhiX RFI DNA with both MfeI and XhoI. 10 μL was collected immediately after the addition of dsDNA (0 min) and the remaining reaction was then incubated at 37°C. 10 μL aliquots were subsequently withdrawn at the indicated timepoints (15, 30, 60, and 120 min). At each timepoint, samples were supplemented with 1 μL of psoralen (200 μg/mL) and subjected to psoralen-UV crosslinking to capture labile DNA structures. Reactions were then deproteinized through addition of 1.8 μL of stop solution (6.6 mg/mL proteinase K and 2.65% SDS) and incubation at 37°C for 30 min. 2.5 μL of loading dye (15% w/v Ficoll, 0.25% Bromophenol blue, 0.25% Xylene cyanol and 20 mM Tris [pH 7.5]) was added to the reactions, and following mixing, 4 μL of the reaction was loaded and resolved in a 0.8% agarose gel in TAE buffer (50 V 120 min). The gel was then stained with SYBR Gold (Thermo Fisher Scientific) and imaged using a LAS4000 mini (GE Healthcare).

In strand exchange assays containing Swi5-Sfr1, 5 μM of Rad51 (wild type, Rad51-E206A, or Rad51-EED) was incubated with 10 μM nt PhiX174 virion DNA (NEB) for 5 min at 37°C in strand exchange buffer (same as above except the total salt concentration was 100 mM KCl). Next, 0.5 μM of Swi5-Sfr1 (or the indicated concentration) was added and following a 5 min incubation at 37°C, 1 μM of RPA was included. After a further 5 min at 37°C, the reaction was initiated with 10 μM nt of PhiX RFI DNA (NEB) linearized with ApaLI and incubated for 120 min at 37°C. Subsequent procedures were carried out as described above.

In strand exchange assays containing Rad54, 7 μM of Rad51 (wild type, Rad51-E206A, or Rad51-EED) was incubated with 7 μM nt PhiX174 virion DNA (NEB) for 8 min at 37°C in strand exchange buffer (same as above except the total salt concentration was 100 mM KCl). Next, 0.7 μM RPA was added and incubation was continued at 37°C for 8 min. The reaction was then initiated with 14 μM nt of the shortened lds (described above) and immediately supplemented with the indicated concentration of Rad54. Incubation was continued for 120 min at 37°C. Subsequent procedures were carried out as described above.

Quantification was performed as previously described (*Haruta et al., 2006*) except that FIJI software was employed (*Schindelin et al., 2012*). Briefly, background was subtracted using the rolling ball method, the signal corresponding to lds, NC/PD, and (JM/1.5) in a given lane was summed and set equal to 100%, and the signal for NC/PD and/or JM was expressed as a percentage of this.

## Immunofluorescence microscopy

Strains were cultured and treated (±UV) exactly as described in *Extraction and detection of cellular proteins*, and nuclear spreads were prepared as described (*Loidl and Lorenz, 2009*; *Argunhan et al., 2017b*) with minor modifications. 10 mL of cell culture (~1×10^8 cells) was harvested by centrifugation (2300 g 2 min), washed with 1 mL of 1 M sorbitol, and then resuspended in 1 mL of lysing solution (1 M sorbitol, 1 mM DTT, 0.2 mg/mL zymolyase, 12 mg/mL lysing enzyme [Sigma-Aldrich L1412]) and incubated at 30°C for 30 min with gentle mixing. Spheroplasts were washed with 1 mL of 1 M sorbitol-1xMES (100 mM 2-[*N*-morpholino] ethanesulfonic acid, 1 mM EDTA, 0.5 mM MgCl$_2$), then resuspended with 100 μL of 1x MES premixed with 350 μL of 3.3% of paraformaldehyde and immediately spread onto glass slides. Once the slides had half-dried (~10 min), they were washed with Photo-Flo 200 Solution (1 mL x2; 146 4510 Kodak) and left to fully air-dry in the dark (~1 hr) before storing at −20°C. For immunostaining, the slides were first washed in a Coplin jar with PBS and blocked using PBS-5% BSA solution for 30 min at room temperature in a moisture chamber. 100 μL of PBS-5% BSA supplemented with affinity purified anti-Rad51 antibody (1:300, provided by Hiroshi Iwasaki) was spread onto the slides with a cover slip and incubated at 4°C overnight in a moisture chamber. Slides were washed with PBS (5 min x3) and subsequent steps were carried out with minimal exposure to ambient light. 100 μL of PBS-5% BSA supplemented with anti-rabbit Alexa fluor 488 antibody (1:300; A11034 Thermo Fisher Scientific) was spread onto the slide with a cover slip, followed by a 3 hr incubation in a moisture chamber at room temperature. Slides were then washed with PBS (5 min x3) and left to air-dry (~30 min). Mounting media (90% glycerol, 0.1xPBS, 1 mg/mL p-Phenylenediamine [Sigma P6001], 1 μg/mL 4′,6-diamidino-2-phenylindole [DAPI; Sigma D9542]) was added in a dropwise fashion onto each slide, spread with a coverslip, and then sealed. Images of nuclei were captured using a wide field fluorescent microscope (Nikon Eclipse 80i) with a 100x objective fitted with an sCMOS camera (C13440 Hamamatsu).

The number of foci for each spread nucleus was quantified using FIJI (*Schindelin et al., 2012*) according to the following procedure. Following the application of a Gaussian blur (sigma = 2) to

the Rad51 (green) channel, the DAPI (blue) and Rad51 channels were subjected to Auto Thresholding (MaxEntropy) and Auto Local Thresholding (Bernsen, radius = 13), respectively. The Rad51 channel was then subjected to the Watershed and Ultimate Points processes, followed by the Find Maxima function. The DAPI channel was subjected to the Analyze Particles process ($\geq$200 pixels$^2$) to add the areas of nuclei to the Region-Of-Interest (ROI) Manager. These ROIs were then overlayed onto the Rad51 channel and the measure function of the ROI Manager was employed. The resulting RawIntDen value for each nucleus was then divided by 255 to yield the number of foci. A graph portraying the results was prepared using GraphPad Prism (version 8), which was also used for statistical analysis (Wilcoxon ranked sum test).

### In vitro IP

Purified proteins (250 nM each) were mixed together on ice in IP buffer (30 mM Tris-Cl [pH 7.5], 150 mM NaCl, 3.5 mM MgCl$_2$, 0.1% igepal CA-630, 0.25 mM TCEP, 1 mM ATP, 5% glycerol; input sample) and incubated at 30℃ for 15 min. Reactions were then incubated on ice for 5 min and supplemented with either anti-FLAG M2 affinity agarose gel (Sigma-Aldrich) for IP of Rad54 or Dynabeads Protein A (ThermoFisher Scientific) preincubated with anti-Sfr1, anti-Rad51, or anti-Rad52 antibodies (*Haruta et al., 2006*; *Kurokawa et al., 2008*). Following mixing at 4℃ for 1.5 hr, the beads were washed with IP buffer (400 µL x1) and eluted in 1x SDS loading buffer (IP sample). Proteins were then separated by SDS-PAGE and detected by immunoblotting with anti-Rad51 (rat 1:10,000; *Argunhan et al., 2020*), anti-Sfr1 (mouse 1:200; *Argunhan et al., 2020*), anti-FLAG (mouse 1:10,000; Sigma-Aldrich F3165), or anti-Rad52 (rabbit 1:10,000; *Kurokawa et al., 2008*) antibodies. FIJI software (*Schindelin et al., 2012*) was used for quantification of in vitro IPs. Briefly, background subtraction was performed by the rolling ball method and the signal for IP'd and co-IP'd protein was quantified. The co-IP'd signal was then divided by the IP'd signal and expressed relative to wild type. Graphs were prepared using GraphPad Prism (version 8).

## Acknowledgements

We thank Yumiko Kurokawa and Yasuto Murayama for help with protein purification, biochemical reconstitutions, and critical reading of the manuscript. We also extend our gratitude to Hiroshi Kimura for support, as well as all members of our laboratory for discussions. This work was supported in part by Grants-in-Aid for Scientific Research on Innovative Areas (15H059749 to HI), for Scientific Research (A) (18H03985 to HI), for Scientific Research (B) (18H02371 to HT), for Young Scientists (B) (17K15061 to BA), for Early-Career Scientists (20K15713 to BA), and a Doctoral Course Fellowship DC2 (17J04051 to NA) from the Japan Society for the Promotion of Science (JSPS).

## Additional information

### Funding

| Funder | Grant reference number | Author |
| --- | --- | --- |
| Japan Society for the Promotion of Science | 15H059749 | Hiroshi Iwasaki |
| Japan Society for the Promotion of Science | 18H03985 | Hiroshi Iwasaki |
| Japan Society for the Promotion of Science | 18H02371 | Hideo Tsubouchi |
| Japan Society for the Promotion of Science | 17K15061 | Bilge Argunhan |
| Japan Society for the Promotion of Science | 20K15713 | Bilge Argunhan |
| Japan Society for the Promotion of Science | 17J04051 | Negar Afshar |

The funders had no role in study design, data collection and interpretation, or the decision to submit the work for publication.

## Author contributions
Negar Afshar, Conceptualization, Data curation, Formal analysis, Funding acquisition, Validation, Investigation, Visualization, Methodology, Writing - original draft, Project administration, Writing - review and editing; Bilge Argunhan, Conceptualization, Resources, Data curation, Software, Formal analysis, Supervision, Funding acquisition, Validation, Investigation, Visualization, Methodology, Writing - original draft, Project administration, Writing - review and editing; Maierdan Palihati, Goki Taniguchi, Validation, Investigation; Hideo Tsubouchi, Formal analysis, Supervision, Funding acquisition, Visualization, Writing - original draft, Project administration, Writing - review and editing; Hiroshi Iwasaki, Conceptualization, Resources, Formal analysis, Supervision, Funding acquisition, Visualization, Writing - original draft, Project administration, Writing - review and editing

## Author ORCIDs
Negar Afshar (iD) http://orcid.org/0000-0002-3448-6710
Bilge Argunhan (iD) https://orcid.org/0000-0002-6023-7654
Hideo Tsubouchi (iD) http://orcid.org/0000-0003-0814-8432
Hiroshi Iwasaki (iD) https://orcid.org/0000-0002-0153-6873

## Decision letter and Author response
Decision letter https://doi.org/10.7554/eLife.64131.sa1
Author response https://doi.org/10.7554/eLife.64131.sa2

# Additional files
## Supplementary files
• Transparent reporting form

## Data availability
All data generated or analysed during this study are included in the manuscript and supporting files. Source data files have been provided for Figures 2-7, Figure 5-figure supplement 1, and Figure 6-figure supplement 1D.

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

# Appendix 1

**Appendix 1—key resources table**

| Reagent type (species) or resource | Designation | Source or reference | Identifiers | Additional information |
|---|---|---|---|---|
| *Schizosaccharomyces pombe; Msmt-0* | *Wild type* | *Akamatsu et al., 2003* | NA310 | *leu-1–32 ura4-D18 his3-D1 arg3-D1* |
| *S. pombe; Msmt-0* | *rad51-E206A-kanMX6* | This study | NA211 | *leu-1–32 ura4-D18 his3-D1 arg3-D1* |
| *S. pombe; Msmt-0* | *rad51::hphMX6* | This study | BA263 | *leu-1–32 ura4-D18 his3-D1 arg3-D1* |
| *S. pombe; Msmt-0* | *rad51-E206A-kanMX6 rad57::hphMX6* | This study | NA253 | *leu-1–32 ura4-D18 his3-D1 arg3-D1* |
| *S. pombe; Msmt-0* | *rad57::hphMX6* | This study | BA268 | *leu-1–32 ura4-D18 his3-D1 arg3-D1* |
| *S. pombe; Msmt-0* | *rad57::hphMX6 rad51::natMX6* | This study | NA383 | *leu-1–32 ura4-D18 his3-D1 arg3-D1* |
| *S. pombe; Msmt-0* | *sfr1::kanMX6* | This study | NA138 | *leu-1–32 ura4-D18 his3-D1 arg3-D1* |
| *S. pombe; Msmt-0* | *sfr1::kanMX6 rad51::hphMX6* | This study | NA233 | *leu-1–32 ura4-D18 his3-D1 arg3-D1* |
| *S. pombe; Msmt-0* | *rad51-E206A-kanMX6 sfr1::kanMX6* | This study | NA236 | *leu-1–32 ura4-D18 his3-D1 arg3-D1* |
| *S. pombe; mat1PD17::LEU2* | *Wild type* | This study | BA396 | *leu-1–32 ura4-D18 his3-D1 arg3-D1* |
| *S. pombe; mat1PD17::LEU2* | *rqh1::natMX6* | This study | NA324 | *leu-1–32 ura4-D18 his3-D1 arg3-D1* |
| *S. pombe; mat1PD17::LEU2* | *rqh1::natMX6 rad51::hphMX6* | This study | NA394 | *leu-1–32 ura4-D18 his3-D1 arg3-D1* |
| *S. pombe; mat1PD17::LEU2* | *rqh1::natMX6 rad52::arg3* | This study | BA546 | *leu-1–32 ura4-D18 his3-D1 arg3-D1* |
| *S. pombe; mat1PD17::LEU2* | *rqh1::natMX6 rad54::ura4* | This study | BA548 | *leu-1–32 ura4-D18 his3-D1 arg3-D1* |
| *S. pombe; mat1PD17::LEU2* | *rqh1::natMX6 rad51-E206A-kanMX6* | This study | NA387 | *leu-1–32 ura4-D18 his3-D1 arg3-D1* |
| *S. pombe; mat1PD17::LEU2* | *rqh1::natMX6 rad57::hphMX6* | This study | NA396 | *leu-1–32 ura4-D18 his3-D1 arg3-D1* |
| *S. pombe; mat1PD17::LEU2* | *rqh1::natMX6 rad57::hphMX6 rad51-E206A-kanMX6* | This study | NA389 | *leu-1–32 ura4-D18 his3-D1 arg3-D1* |
| *S. pombe; mat1PD17::LEU2* | *rqh1::natMX6 sfr1::kanMX6* | This study | NA386 | *leu-1–32 ura4-D18 his3-D1 arg3-D1* |
| *S. pombe; mat1PD17::LEU2* | *rqh1::natMX6 sfr1::kanMX6 rad51-E206A-kanMX6* | This study | NA398 | *leu-1–32 ura4-D18 his3-D1 arg3-D1* |
| *S. pombe; mat1PD17::LEU2* | *rqh1::natMX6 rad57::hphMX6 sfr1::kanMX6* | This study | NA391 | *leu-1–32 ura4-D18 his3-D1 arg3-D1* |
| *S. pombe; Msmt-0* | *rad52::arg3* | Hiroshi Iwasaki (lab stock) | BA481 | *leu-1–32 ura4-D18 his3-D1 arg3-D1* |
| *S. pombe; Msmt-0* | *rad54::ura4* | Hiroshi Iwasaki (lab stock) | BA518 | *leu-1–32 ura4-D18 his3-D1 arg3-D1* |

*Continued on next page*

*Appendix 1—key resources table continued*

| Reagent type (species) or resource | Designation | Source or reference | Identifiers | Additional information |
|---|---|---|---|---|
| S. pombe; Msmt-0 | rad51-E205A-E206A-D209A-kanMX6 | This study | BA499 | leu-1–32 ura4-D18 his3-D1 arg3-D1 |
| S. pombe; Msmt-0 | rad51+-kanMX6 | This study | NA209 | leu-1–32 ura4-D18 his3-D1 arg3-D1 |
| S. pombe; Msmt-0 | rad51-E205A-kanMX6 | This study | BA445 | leu-1–32 ura4-D18 his3-D1 arg3-D1 |
| S. pombe; Msmt-0 | rad51-D209A-kanMX6 | This study | BA493 | leu-1–32 ura4-D18 his3-D1 arg3-D1 |
| S. pombe; Msmt-0 | rad51-E205A-E206A-kanMX6 | This study | BA449 | leu-1–32 ura4-D18 his3-D1 arg3-D1 |
| S. pombe; Msmt-0 | rad55-12xV5-natMX6 | This study | BA172 | leu-1–32 ura4-D18 his3-D1 arg3-D1 |
| S. pombe; Msmt-0 | rad55-12xV5-natMX6 rad51-E206A-kanMX6 | This study | NA238 | leu-1–32 ura4-D18 his3-D1 arg3-D1 |
| S. pombe; Msmt-0 | rad55-12xV5-natMX6 sfr1::kanMX6 | This study | BA170 | leu-1–32 ura4-D18 his3-D1 arg3-D1 |
| S. pombe; Msmt-0 | rad55-12xV5-natMX6 rad51-E206A-kanMX6 sfr1::kanMX6 | This study | NA243 | leu-1–32 ura4-D18 his3-D1 arg3-D1 |
| S. pombe; Msmt-0 | rad55-12xV5-natMX6 rad51-E205A-E206A-kanMX6 | This study | BA529 | leu-1–32 ura4-D18 his3-D1 arg3-D1 |
| S. pombe; Msmt-0 | rad55-12xV5-natMX6 rad51 D209A-kanMX6 | This study | BA521 | leu-1–32 ura4-D18 his3-D1 arg3-D1 |
| S. pombe; Msmt-0 | rad55-12xV5-natMX6 rad51-E205A-E206A-D209A-kanMX6 | This study | BA523 | leu-1–32 ura4-D18 his3-D1 arg3-D1 |
| S. pombe; Msmt-0 | rad57::hphMX6 rad51+-kanMX6 | This study | NA249 | leu-1–32 ura4-D18 his3-D1 arg3-D1 |
| S. pombe; Msmt-0 | rad57::hphMX6 rad51-E205A-kanMX6 | This study | BA443 | leu-1–32 ura4-D18 his3-D1 arg3-D1 |
| S. pombe; Msmt-0 | rad57::hphMX6 rad51-D209A-kanMX6 | This study | BA491 | leu-1–32 ura4-D18 his3-D1 arg3-D1 |
| S. pombe; Msmt-0 | rad57::hphMX6 rad51-E205A-E206A-D209A-kanMX6 | This study | BA501 | leu-1–32 ura4-D18 his3-D1 arg3-D1 |
| S. pombe; Msmt-0 | rad57::hphMX6 rad51-E205A-E206A-kanMX6 | This study | BA447 | leu-1–32 ura4-D18 his3-D1 arg3-D1 |
| S. pombe; Msmt-0 | sfr1::kanMX6 rad51+-kanMX6 | This study | NA222 | leu-1–32 ura4-D18 his3-D1 arg3-D1 |
| S. pombe; Msmt-0 | sfr1::natMX6 rad51-E205A-kanMX6 | This study | BA461 | leu-1–32 ura4-D18 his3-D1 arg3-D1 |
| S. pombe; Msmt-0 | sfr1::natMX6 rad51-E206A-kanMX6 | This study | BA476 | leu-1–32 ura4-D18 his3-D1 arg3-D1 |
| S. pombe; Msmt-0 | sfr1::natMX6 rad51-D209A-kanMX6 | This study | BA485 | leu-1–32 ura4-D18 his3-D1 arg3-D1 |
| S. pombe; Msmt-0 | sfr1::nathMX6 rad51-E205A-E206A-kanMX6 | This study | BA463 | leu-1–32 ura4-D18 his3-D1 arg3-D1 |

*Appendix 1—key resources table continued*

| Reagent type (species) or resource | Designation | Source or reference | Identifiers | Additional information |
|---|---|---|---|---|
| *S. pombe; Msmt-0* | *sfr1::natMX6 rad51-E205A-E206A-D209A-kanMX6* | This study | BA497 | *leu-1–32 ura4-D18 his3-D1 arg3-D1* |
| *S. pombe; Msmt-0* | *rad55::natMX6* | *Argunhan et al., 2020* | BA150 | *leu-1–32 ura4-D18 his3-D1 arg3-D1* |
| *S. pombe; Msmt-0* | *rad55-12xV5-natMX6* | This study | BA165 | *leu-1–32 ura4-D18 his3-D1 arg3-D1* |
| *S. pombe; Msmt-0* | *rad55::arg3 sfr1::kanMX6* | *Argunhan et al., 2020* | BA126 | *leu-1–32 ura4-D18 his3-D1 arg3-D1* |
| *S. pombe; Msmt-0* | *rad51-E206K-kanMX6* | This study | BA552 | *leu-1–32 ura4-D18 his3-D1 arg3-D1* |
| Antibody | Rat polyclonal anti-Rad51 | *Argunhan et al., 2020* | | 1:10,000 |
| Antibody | Mouse monoclonal anti-tubulin | Sigma-Aldrich | T5168 | 1:10,000 |
| Antibody | Mouse monoclonal anti-V5 | Bio-Rad | MCA1360 | 1:10,000 4 µL / IP |
| Antibody | Rabbit polyclonal anti-Rad51 | *Haruta et al., 2006* | | 10 µL / IP |
| Antibody | ChromPure Human IgG, whole molecule polyclonal | Jackson Immunoresearch Laboratories | 009-000-003 | 0.336 µL / IP |
| Antibody | Rabbit monoclonal Alexa fluor 488 | Thermo Fisher Scientific | A11034 | 1:300 |
| Antibody | Mouse monoclonal anti-Sfr1 | *Argunhan et al., 2020* | | 1:200 |
| Antibody | Mouse monoclonal anti-FLAG-tag | Sigma-Aldrich | F3165 | 1:10,000 |
| Antibody | Rabbit polyclonal anti-Rad52 | *Kurokawa et al., 2008* | | 1:10,000 |
| Antibody | Anti-mouse IgG (HRP-conjugated) | GE Healthcare | NA931 | 1:5000 |
| Antibody | Anti-rabbit IgG (HRP-conjugated) | GE Healthcare | NA934 | 1:5000 |
| Antibody | Anti-rat IgG (HRP-conjugated) | Jackson Immunoresearch Laboratories | 712-035-153 | 1:10,000 |
| Recombinant DNA reagent | PhiX174 virion DNA (ssDNA plasmid) | NEB | N3023L | |
| Recombinant DNA reagent | PhiX174 RF I DNA (dsDNA plasmid) | NEB | N3021L | |
| Peptide, recombinant protein | *S. pombe* Rad51 | *Kurokawa et al., 2008* | | |
| Peptide, recombinant protein | *S. pombe* Rad51-E206A | This study | | |
| Peptide, recombinant protein | *S. pombe* Rad51 E205A-E206A-D209A | This study | | |
| Peptide, recombinant protein | *S. pombe* RPA | *Haruta et al., 2006* | | |

*Continued on next page*

*Appendix 1—key resources table continued*

| Reagent type (species) or resource | Designation | Source or reference | Identifiers | Additional information |
|---|---|---|---|---|
| Peptide, recombinant protein | *S. pombe* Swi5-Sfr1 | **Haruta et al., 2006** | | |
| Peptide, recombinant protein | *S. pombe* Rad52 | **Kurokawa et al., 2008** | | |
| Peptide, recombinant protein | *S. pombe* Rad54 | This study | | |
| Commercial assay or kit | Amicon Ultra-15, 10K MWCO | Merck | UFC901096 | |
| Commercial assay or kit | Dynabeads Protein A | ThermoFisher | 10002D | |
| Commercial assay or kit | ImmunoStar Zeta chemiluminescence solution | FujiFilm Wako | 297–72403 | |
| Commercial assay or kit | Malachite Green Phosphate Assay Kit | BioAssay Systems | POMG-25H | |
| Commercial assay or kit | TurboNuclease | Accelagen | N0103P | |
| Commercial assay or kit | Lysing enzyme | Sigma-Aldrich | L1412 | |
| Commercial assay or kit | In-Fusion HD Cloning | TaKaRa | 639650 | |
| Chemical compound, drug | ATP | Sigma-Aldrich | A2383 | |
| Chemical compound, drug | Phosphocreatine di(tris) salt | Sigma-Aldrich | P1937 | |
| Chemical compound, drug | Bio-Safe Coomassie Stain | Bio-Rad | 1610786 | |
| Chemical compound, drug | Proteinase K | TaKaRa | 9034 | |
| Chemical compound, drug | Creatine Kinase | Sigma-Aldrich | 10127566001 | |
| Sequenced-based reagent | 398_F | This study | PCR primers | CCTCTAGAAATAATTTTG TTTAACTTTA AGAAGGAGATATACATA |
| Sequenced-based reagent | 400_R | This study | PCR primers | ATCCAAAACTGCCTCACCA TTTAAGCCATA |
| Sequenced-based reagent | 399_F | This study | PCR primers | AATGGTGAGGCAGTTTTGGA TAACGTTGCA |
| Sequenced-based reagent | 405_R | This study | PCR primers | CTTTGTTAGCAGCCGGATCC |
| Sequenced-based reagent | 350_F | This study | PCR primers | GATTGGAAATCGTAAGGA TCCGCGAATTTC TTATGATTTATGATTTTTATT |
| Sequenced-based reagent | 351_R | This study | PCR primers | ATAGTAAGGAGTCCTCAGTA TAGCGACCAGCATTC |
| Sequenced-based reagent | 409_F | This study | PCR primers | CGGTATCGATAAGCTTGATA TCCATAA GACTGAGGCAGAGGT |
| Sequenced-based reagent | 412_R | This study | PCR primers | CTCTGTATCTGCCATTATAAC TTGTTA AGCACGAAATTATCAC |

*Continued on next page*

*Appendix 1—key resources table continued*

| Reagent type (species) or resource | Designation | Source or reference | Identifiers | Additional information |
|---|---|---|---|---|
| Sequenced-based reagent | 417_F | This study | PCR primers | CTGGTCGCTATACTGATTA TTTTAGTATCGT TTCATTTTTATTTATTA TTTTGCA |
| Sequenced-based reagent | 419_R | This study | PCR primers | GCGGTGGCGGCCGCTC TAGAGCATGCTTGGAAGGC TTT |
| Sequenced-based reagent | 200_F | This study | PCR primers | CTACTGGTATTCAGGATTA TCAAAGTATTCCTACC AATAGCTCACAACGACG TAAGAGATCCATTTTGGA ATGTGAGTCCCGGA TCCCCGGGTTAATTAA |
| Sequenced-based reagent | 201_R | This study | PCR primers | TAGGTATAATAAATTAATAGA TATGGGCAAAACAAC CATCACTATGCTAAAAAA TTCGTAACTGAAGCCACCA TTTTTACGAATTCGAGCTCG TTTAAAC |
| Software, algorithm | FIJI | *Schindelin et al., 2012* | | |
| Software, algorithm | Prism version 8 | GraphPad | | |
| Other | DAPI stain | Sigma-Aldrich | D9542 | (1 mg/mL) |

