## [Decision Letter]

**Acceptance summary:**

This is an excellent contribution that identified an acidic patch as a key interaction site on fission yeast Rad51 that mediates the interactions with Rad52, Rad54, and the Rad55-Rad57 paralog complex. The combined genetic, biochemical, and cytological analysis of the mutants is both insightful and compelling. The authors can exclude a potential artifact that the interaction mutants affect protein structure/function in a more trivial manner. The insights have significant impact on the thinking how the Rad51-ssDNA filament coordinates the interactions with its key interaction partners, Rad52, Rad54, and Rad55-Rad57. In the revision the authors addressed all points in the critique and further strengthened an already strong contribution.

**Decision letter after peer review:**

Thank you for submitting your article "A novel motif of Rad51 serves as an interaction hub for recombination auxiliary factors" for consideration by *eLife*. Your article has been reviewed by three peer reviewers, including Wolf-Dietrich Heyer as the Reviewing Editor and Reviewer #1, and the evaluation has been overseen by Jessica Tyler as the Senior Editor.

The reviewers have discussed the reviews with one another and the Reviewing Editor has drafted this decision to help you prepare a revised submission.

Summary:

This is an excellent contribution that identified an acidic patch as a key interaction site on fission yeast Rad51 that mediates the interactions with Rad52, Rad54, and the Rad55-Rad57 paralog complex. The study succeeds of identifying Rad51 interaction sites with these factors, where earlier studies in budding yeast failed. The combined genetic, biochemical, and cytological analysis of the mutants is both insightful and compelling. The authors can exclude a potential artifact that the interaction mutants affect protein structure/function in a more trivial manner. The insights will have significant impact on the thinking how the Rad51-ssDNA filament coordinates the interactions with its key interaction partners, Rad52, Rad54, and Rad55-Rad57. The manuscript stands on its own without further experimentation. Acceptance depends on the points listed below, which ask for clarifications and explanations. The authors may want to consider two points (#1, 2) about experimental controls that can be easily addressed.

Essential revisions:

1) Potential experimental addition: Figure 7A: Please show the immunoblot comparing wild type with *rad52Δ* to prove that the labeled band is indeed Rad52 protein.

2) Potential experimental addition: Figure 7C: Is the result of the reduced *rad51-EED:Rad54* reproducible and significant? Quantitation of several independent experiments would be better here.

3) Are the molecular "models" (e.g., Figure 1) based on physical determination (X-ray crystallography or cryoEM, for example), or are they computational models based on other known structures? This should be clarified.

4) The use of *rqh1Δ* in Figure 2D should be explained. Is it more sensitive to HU than wt, and if so, why? Why does *rad51Δ* weakly, and *rad51-E206A* or *sfr1Δ* strongly, suppress that sensitivity? While this assay shows the deduced interactions of Rad51-E206A with Rad57 and Sfr1, knowing the basis of this assay would help.

5) It's unclear what part of Rad55-57 etc. binds to the PAP. Is there a "basic patch" on it? Is the FxxA motif thought to bind the PAP? If so, how? (I'd expect FxxA to bind a hydrophobic patch.) Similarly for Rad52 and Rad54.

6) Further explanation of the strand exchange assays (Figures 3, 6, and 7) would help. What is limiting in these assays (e.g., subsection “Rad51-E206A retains normal recombinase activity and can be stimulated by Swi5-Sfr1”)? Why do the results for Rad51-E206A in Figure 3E differ from those in Figure 3F? Why is the concentration of Rad51 (etc.) different in panels E and F of Figure 3?

7) Is the anti-Rad51 antibody (subsection “The PAP is crucial for the recruitment of Rad51 to DNA damage sites”) mono- or polyclonal? Does it bind the mutant Rad51-EED protein? Without knowing it binds the mutant, interpreting the data is hard.

8) In the subsection “Rad51-EED retains intrinsic recombinase activity and can be stimulated by Swi5-Sfr1”, does Rad51-E206K interfere with Swi5-Sfr1 binding or action? Perhaps PAP is important for that interaction, and this stronger mutation might block it.

9) In the subsection “The PAP plays a central role in the interaction of Rad51 with recombination auxiliary factors”, it says PAP is close to the Phe of FxxA, but later says 13 – 18 Angstroms. This does not seem "close," because it is about 1/3 of the distance across a Rad51 molecule. Is there evidence that amino acids interact over such a distance?

10) Which nuclease was used in the experiments in Figures 2, 4, and 7?

---

## [Author Response]

Essential revisions:1) Potential experimental addition: Figure 7A: Please show the immunoblot comparing wild type with rad52Δ to prove that the labeled band is indeed Rad52 protein.

This experiment has now been included as Figure 7—figure supplement 1A. The band that migrates slightly faster than the 60 kDa size marker is the only band that is not observed in the *rad52∆* strain. The positions of relevant size markers have also been indicated in Figure 7A to allow a more meaningful comparison with this control experiment. A brief description of this has been added to the legend for Figure 7. Please note that the intensities of the nonspecific bands relative to Rad52 are slightly different in Figure 7A and Figure 7—figure supplement 1A. This is because the former is a sample of soluble protein extract whereas the latter is the whole-cell extract prepared by TCA precipitation. We have also seen that the relative intensities of these bands can vary depending on the extent of blocking.

2) Potential experimental addition: Figure 7C: Is the result of the reduced rad51-EED:Rad54 reproducible and significant? Quantitation of several independent experiments would be better here.

The Rad54-Rad51 co-IP experiment has now been performed in triplicate and quantified. As shown in Figure 7C, the reduced interaction observed for Rad51-EED is reproducible and statistically significant. By contrast, the interaction of Rad51-E206A with Rad54 is comparable to wild type. The description of these results has been updated (subsection “The PAP is important for the interaction of Rad51 with both Rad52 and Rad54”).

3) Are the molecular "models" (e.g., Figure 1) based on physical determination (X-ray crystallography or cryoEM, for example), or are they computational models based on other known structures? This should be clarified.

All molecular models of *Sp*Rad51 are homology (i.e., computational) models. This has now been clearly stated in the text (subsection “E205, E206, and D209 comprise a protruding acidic patch (PAP) on the exterior of the Rad51 presynaptic filament”). In addition, all figures containing molecular models now have a small annotation specifying whether the model is a homology model, derived from cryo-EM, or derived from X-ray crystallography. Further details are also available in the Materials and methods sections.

4) The use of rqh1Δ in Figure 2D should be explained. Is it more sensitive to HU than wt, and if so, why? Why does rad51Δ weakly, and rad51-E206A or sfr1Δ strongly, suppress that sensitivity? While this assay shows the deduced interactions of Rad51-E206A with Rad57 and Sfr1, knowing the basis of this assay would help.

Thank you for bringing this oversight to our attention. We have now included a more detailed explanation of the basis behind the *rqh1∆* experiment, including two additional references (subsection “Rad51-E206A is specifically defective in the interaction with Rad55-Rad57”). Briefly, toxic HR intermediates can accumulate in the absence of Rqh1. Mutations that impair HR suppress the production of these intermediates to differing extents. Those that mildly impair HR (i.e., *rad57∆, sfr1∆, rad51-E206A*) strongly suppress the sensitivity to HU by reducing the formation of these toxic intermediates. While mutations that severely impair HR (i.e., *rad51∆, rad52∆, rad54∆, rad57∆ sfr1∆*) also reduce the formation of these toxic intermediates, they also abolish (or nearly abolish) HR itself, leading to HU sensitivity and reduced suppression of *rqh1∆*.

5) It's unclear what part of Rad55-57 etc. binds to the PAP. Is there a "basic patch" on it? Is the FxxA motif thought to bind the PAP? If so, how? (I'd expect FxxA to bind a hydrophobic patch.) Similarly for Rad52 and Rad54.

Evidently, the explanation of our model was inadequate. We apologise for the confusion this may have caused. We completely agree that the FxxA motif of each auxiliary factor would be expected to interact with a hydrophobic region. In fact, we expect that FxxA would insert into the same hydrophobic pocket employed by Rad51 itself. We did not mean to suggest that the PAP of Rad51 would bind to the FxxA motif of each auxiliary factor. Rather, we propose that a basic patch close to the FxxA motif of each auxiliary factor would bind to the PAP through electrostatic interactions, which would then facilitate the insertion of the FxxA motif of an auxiliary factor into the hydrophobic pocket of Rad51. This is also related to point (9). We employed the Rad51 filament model—in which the FxxA motif of one Rad51 monomer inserts into the hydrophobic pocket of the adjacent monomer—as a proxy for how we envision an auxiliary factor might bind to Rad51 via Rad51 mimicry (Figure 7E). When we stated that the PAP is in close proximity to the FxxA motif (13-18 Å), we did not mean that the PAP could directly interact with FxxA over such distances. Instead, we infer that the PAP is well-placed to bind positively charged regions that are close in 3D space to the FxxA motif of auxiliary factors. We speculate that the PAP functions as a landing pad, which perhaps facilitates the orientation of auxiliary factors or functions as a scaffold to promote FxxA insertion. We have now revised the relevant section of the Discussion and hope this has improved clarity.

Related to this, we also included a reference to a highly complementary paper with *S. cerevisiae* proteins showing that mutation of the Phe residue in the FxxA motif of Rad52 disrupts the interaction with Rad51 (Kagawa et al., 2014).

6) Further explanation of the strand exchange assays (Figures 3, 6, and 7) would help. What is limiting in these assays (e.g., subsection “Rad51-E206A retains normal recombinase activity and can be stimulated by Swi5-Sfr1”)? Why do the results for Rad51-E206A in Figure 3E differ from those in Figure 3F? Why is the concentration of Rad51 (etc.) different in panels E and F of Figure 3?

For each assay, we have now clearly stated that the conditions may differ between figures and have included an explanation of why we employed each condition (Results). Unlike *Sc*Rad51, *Sp*Rad51 does not efficiently promote JM/NC formation in the three-strand exchange assay even when the order-of-addition is favourable, making it difficult to compare the intrinsic strand exchange activity of wild-type Rad51 with mutants. We were able to partially overcome this problem by increasing the concentration of substrate (3x ssDNA, 3x Rad51, 2x dsDNA, x is relative to our standard reaction condition [Haruta et al., 2006; Kurokawa et al., 2008]; Figure 3E). We were later able to circumvent this problem entirely by utilizing a shorter dsDNA substrate (Figure 6B). In order to more clearly examine the effect of Swi5-Sfr1 (Figures 3F, 6D), we revert to the standard reaction condition in which Rad51 alone is unable to form JM/NC, as this provides a cleaner comparison between ± Swi5-Sfr1. For Rad54, we again employ a condition where wild-type Rad51 alone does not show any activity (Figure 7D). However, in this case, we modify the concentrations of substrate slightly since we have to employ the short dsDNA substrate (the full-length substrate yields high-molecular weight species that likely arise from a single ssDNA molecule engaging multiple dsDNA molecules, as previously reported with *S. cerevisiae* Rad54 [Petukhova et al., 1998; Solinger et al., 2001]).

7) Is the anti-Rad51 antibody (subsection “The PAP is crucial for the recruitment of Rad51 to DNA damage sites”) mono- or polyclonal? Does it bind the mutant Rad51-EED protein? Without knowing it binds the mutant, interpreting the data is hard.

The anti-Rad51 antibody is polyclonal, thus it is highly unlikely that three amino acid substitutions would significantly impact its affinity for Rad51. Consistent with this notion, we did not see a difference in the ability of the antibody to recognize purified Rad51-EED in immunoblotting experiments (e.g., Figure 6C).

8) In the subsection “Rad51-EED retains intrinsic recombinase activity and can be stimulated by Swi5-Sfr1”, does Rad51-E206K interfere with Swi5-Sfr1 binding or action? Perhaps PAP is important for that interaction, and this stronger mutation might block it.

We purified Rad51-E206K and examined the physical interaction with purified Swi5-Sfr1. Similar to Rad51-EED, we saw a reproducible reduction in the co-IP of Rad51-E206K with Swi5-Sfr1, but this reduction was relatively subtle (Figure 6—figure supplement 1D). A description of this result has been included in the revised manuscript (subsection “Rad51-EED retains intrinsic recombinase activity and can be stimulated by Swi5-Sfr1”).

9) In the subsection “The PAP plays a central role in the interaction of Rad51 with recombination auxiliary factors”, it says PAP is close to the Phe of FxxA, but later says 13 – 18 Angstroms. This does not seem "close," because it is about 1/3 of the distance across a Rad51 molecule. Is there evidence that amino acids interact over such a distance?

Please see our response to point (5) above.

10) Which nuclease was used in the experiments in Figures 2, 4, and 7?

The commercial name of the nuclease is described in the Materials and methods (TurboNuclease from Accelagen). This is a benzonase-like endonuclease that non-specifically targets both ssDNA and dsDNA. The manufacturer does not call it benzonase, probably for proprietary reasons, although it is essentially marketed as a cheaper alternative to benzonase. We have added the “benzonase-like” descriptor to each figure legend containing in vivo IP results.